

# Comparison of dealiasing schemes in large-eddy simulation of neutrally-stratified atmospheric boundary-layer type flows

Fabien Margairaz[1], Marco G. Giometto[2], Marc B. Parlange[2], and Marc Calaf[1]

[1]University of Utah, Department of Mechanical Engineering, 1495 E 100 S, Salt Lake City, Utah 84112, USA
[2]University of British Columbia, Faculty of Applied Science, 2332 Main Mall, Vancouver BC V6T 1Z4, Canada

*Correspondence to:* Fabien Margairaz (fabien.margairaz@utah.edu)

**Abstract.** Three dealiasing schemes for large-eddy simulation of turbulent flows are inter-compared for the canonical case of pressure-drive atmospheric boundary-layer type flows. Aliasing errors arise in the multiplication of partial sums, such as those encountered when integrating the non-linear terms of the Navier-Stokes equations in spectral methods (Fourier or polynomial discrete series), and are detrimental to the accuracy of the numerical solution. This is of special relevance when using high-order schemes. In this work, a performance/cost analysis is developed for three well-accepted approaches: the exact 3/2 rule, the Fourier truncation method, and a high order Fourier smoothing method. Tests are performed within a newly developed mixed pseudo-spectral collocation - finite differences large-eddy simulation code, parallelized using a two-dimensional pencil decomposition. The static Smagorinsky eddy-viscosity model with wall damping of the model coefficient is used. A series of simulations are performed at varying resolution and key flow statistics are inter-compared among the considered dealiasing schemes. The numerical results validate the numerical performance predicted by theory when using the Fourier truncation and Fourier smoothing methods. In terms of turbulence statistics, the Fourier Truncation method proves to be over-dissipative when compared against the Fourier Smoothing method and the traditional 3/2-rule, leading to an enhanced horizontal integrated mass flux and to higher dispersive momentum fluxes. Its use in large-eddy simulation of atmospheric boundary-layer type flows is therefore not recommended. Conversely, the Fourier Smoothing method yields accurate flow statistics, comparable to those resulting from the application of the 3/2 rule, with a significant reduction in computational cost, which makes it a convenient alternative for use in the studies related to the atmospheric boundary layer.

## 1 Introduction

The past decades have seen significant progress in computer hardware in remarkable agreement with Moore's law, which states that the number of nodes in the discretization grids double every eighteen months (Moore, 1965; Voller and Porté-Agel, 2002; Takahashi, 2005). A comparable progress has been made in software development, with the rise of new branches in numerical analysis like reduced order modeling (Burkardt et al., 2006) and uncertainty quantification (Najm, 2009), as well as





the development of highly efficient numerical algorithms and computing frameworks like Isogeometric Analysis (Hsu et al., 2011) or GPU-computing (Hamada et al., 2009; Bernaschi et al., 2010). With increasing computer power the range of scales and applications in computational fluid dynamics (CFD) has significantly broadened, allowing to describe at an unprecedented level of detail complex flow systems such as fluid-structure interaction (Hughes et al., 2005; Bernaschi et al., 2010; Takizawa

and Tezduyar, 2011), land-atmosphere exchange of scalars, momentum and mass (Moeng, 1984; Bou-Zeid et al., 2004; Tseng et al., 2006; Calaf et al., 2010; Anderson et al., 2012; Giometto et al., 2016), weather research and forecasting (Skamarock et al., 2008), micro-fluidics (Wörner, 2012), and canonical wall bounded flows (Schlatter and Örlü, 2012), to name but a few. Despite this progress, very high Reynolds number flows – such as those characterizing the atmospheric boundary layer (ABL) – still remain challenging with the cost of full resolution numerical simulations remaining out of reach. As a result,

methods that aim at reducing computational requirements while preserving numerical accuracy are still of great interest. The Fourier-based pseudo-spectral collocation method (Orszag, 1970; Orszag and Pao, 1975; Canuto et al., 2006) remains the preferred "work-horse" in simulations of wall-bounded flows over horizontally periodic regular domains. This is often used in conjunction with a centered finite-difference or Chebychev polynomial expansions in the vertical direction (Shah and Bou-Zeid, 2014; Moeng and Sullivan, 2015). The main strength of such an approach is the high-order of accuracy of the Fourier

partial sum representation, coupled with the intrinsic efficiency of the fast-Fourier-transform algorithm (Cooley and Tukey, 1965; Frigo and Johnson, 2005). In such algorithms the leading order error term is represented by the aliasing that arises when evaluating the quadratic non-linear term (convective fluxes of momentum). This was first discovered in the early works of Orszag and Patterson (Orszag, 1971; Patterson, 1971) which also set a cornerstone in the treatment and removal of aliasing errors in pseudo-spectral collocation methods. Aliasing errors can severely deteriorate the quality of the solution, as exemplified

by the large body of literature that has dealt with the topic. In Horiuti (1987) and Moin and Kim (1982), it was shown how the energy-conserving rotational form of the Large-Eddy Simulation (LES) equations performed poorly without dealiasing, and that proper removal of such error significantly improved the accuracy of the solution on statistics like the flow turbulent shear stress, turbulence intensities and two point correlations. As shown in Moser et al. (1983); Zang (1991); Kravchenko and Moin (1997), aliasing errors do not alter the energy conservation properties of the rotational form of the LES equations, but

the additional dissipation that is introduced makes the flow prone to laminarization. Dealiasing is hence required in order to accurately resolve turbulent flows with a well developed inertial subrange, such as ABL flows for instance. However, the classic (exact) dealiasing methods developed in (Patterson, 1971) based on padding and truncation (the 3/2 rule) or on the phase-shift technique, have proven to be computationally expensive, and are one of the most costly module for momentum integration in high resolution simulations, as it will be shown later in this work. For example, in simulations with Cartesian discretization,

where $N$ is the number of collocation nodes in each of the three coordinate directions, the 3/2 rule requires to expand the number of nodes to $3/2 \times N$, and the phase-shift method needs grids with $2 \times N$ nodes. As a result, due to the non-linear dependence on N, the computational burden introduced by such methods is high, and increases with increasing grid resolution, thus severely limiting computational performances in large scale models. This has motivated efforts towards the development of approximate yet computationally efficient dealiasing schemes, such as the Fourier truncation (FT) method (Orszag, 1971;

Moeng, 1984; Moeng and Wyngaard, 1988; Albertson, 1996), the Fourier smoothing (FS) method (Hou and Li, 2007), and the





more recent implicit dealiasing of Bowman et al. (Bowman and Roberts, 2011). Details on the FT and the FS techniques are provided in the following section. Limits and merits of the different dealiasing techniques have been extensively discussed in the past decades within the turbulent flow framework (Moser et al., 1983; Zang, 1991).

In this work we provide a cost-benefit analysis for the FT and FS dealiasing schemes in comparison to the exact 3/2-rule, via a set of LES of fully developed ABL type flows and with a corresponding comparison on the effect in turbulent flow statistics and topology. Simulations and benchmark analyses are performed using a mixed pseudo-spectral finite difference code parallelized using a pencil decomposition technique based on the 2DECOMP&FFT library (Li and Laizet, 2010). Results of this work are of prime interest to the environmental fluid community (e.g. ABL community) because they can help improve the numerical performance of some of the numerical approaches used. An overview on the different dealiasing methods is provided in section 2. Section 3 briefly presents the LES platform with important benchmark results. The computational cost analysis and flow statistics obtained with the different dealiasing schemes are later discussed in section 4. Finally, the conclusions are presented in section 5.

## 2  Dealiasing methods

Aliasing errors result from representing the product of two or more variables by $N$ wave-numbers, when each one of the variables is itself represented by a finite sum of $N$ terms (Canuto et al., 2006), here $N$ is assumed even. Such is the case for example when treating the non-linear advection term in the Navier-Stokes (NS) equations. Let $f$ and $g$ be two smooth functions with the corresponding discrete Fourier transforms expressed as,

$$f(x) = \sum_{k=-N/2}^{N/2-1} \hat{f}_k e^{ikx} \quad \text{and} \quad g(x) = \sum_{m=-N/2}^{N/2-1} \hat{g}_k e^{ikx}, \tag{1}$$

with $\hat{f}_k$ and $\hat{g}_k$ being the amplitudes of the $k$-th Fourier mode of $f$ and $g$. The product of the two functions is hence given by

$$h(x) = f(x)g(x) = \sum_{m=-N/2}^{N/2-1} \hat{f}_m e^{imx} \sum_{n=-N/2}^{N/2-1} \hat{g}_n e^{inx} = \sum_{k=-N}^{N-1} \hat{h}_k e^{ikx}, \tag{2}$$

with

$$\hat{h}_k = \sum_{m+n=k} \hat{f}_n \hat{g}_m \quad \text{and} \quad |m|, |n| \leq N/2. \tag{3}$$

Note that the corresponding expression for the Fourier transform of the product $h$ (result of the convolution of $f$ with $g$) requires $2N$ modes. Therefore, the exact computation of the product represents a major numerical cost. Traditionally the convolution of the two functions $f$ and $g$ is made with only $N$ Fourier modes,

$$h(x) = \sum_{k=-N/2}^{N/2-1} \tilde{h}_k e^{ikx}. \tag{4}$$





As a result, the energy contained within the remaining $N+1$ to $2N$ modes folds back on the first $N$ modes, and the amplitude of the first $N$ modes ($\tilde{h}_k$) is aliased. This can be related to the exact amplitude $\hat{h}_k$ as,

$$\hat{h}_k = \sum_{m+n=k} \hat{f}_n \hat{g}_m + \sum_{m+n=k\pm N} \hat{f}_n \hat{g}_m = \tilde{h}_k + \sum_{m+n=k\pm N} \hat{f}_n \hat{g}_m, \tag{5}$$

with

$$\tilde{h}_k = \sum_{m+n=k} \hat{f}_n \hat{g}_m \quad \text{and} \; -N/2 \leq k \leq N/2 - 1, \tag{6}$$

such that the second term contains the aliasing errors on the $k$-th mode. Aliasing errors propagate in the solution of the differential equation and can induce large errors. For the pseudo-spectral methods, the truncation and aliasing errors affect both the accuracy and the stability of the numerical solution (see (Canuto et al., 2006) chap. 3 and (Canuto et al., 2007) chap. 3 for detailed discussion). Traditionally, the aliasing errors are treated using one of the two methods discussed below.

The 3/2 rule method is based on the so-called padding and truncation technique, where the Fourier partial sums are zero-padded in Fourier space by half the available modes (from $N$ to $3/2N$), inverse-transformed to physical space before multiplication, multiplied, and then truncated back to the original variable size ($N$). This method fully removes aliasing errors. However, the high computational cost related to the inverse transform operation discourages its use in large scale simulations as the fast Fourier transform (FFT) algorithm has an operational complexity of $N \log_2(N)$. Thus, counting the number of FFT and multiplications, the operation count of the 3/2-rule applied to dealias the product of two vectors of N components becomes $(45/4)N \log_2(3N/2)$ (Canuto et al., 2006). An alternative method is the so-called Phase Shift method which consists in shifting the grid of one of the variables in physical space. Given the appropriate shift, the aliasing errors are eliminated naturally in the evaluation of the convolution sum. This method, however, has a cost equal to $15N \log_2(N)$ (Canuto et al., 2006), which is even greater than the 3/2-rule (Patterson, 1971; Orszag, 1972), thus discouraging its use in most practical situations. The discussion above concerns one dimensional problems but the expansion to higher dimensional problems is straightforward (Iovieno et al., 2001; Canuto et al., 2006). Although, this method provides the full dealiasing of the non-linear term, the cost of expanding the number of Fourier modes by a factor of $3/2$ is a computationally expensive endeavour, especially with the progressive increasing size of numerical grids. To reduce the numerical burden, two additional methods were proposed in the past for treating the aliasing errors: the FS method (Hou and Li, 2007), and the FT method (Orszag, 1971; Albertson and Parlange, 1999). In both methods, a set of high-wave-number Fourier amplitudes are multiplied by a test function $\hat{u}_k^* = f(k)u_k$, to avoid expansions to larger number of modes. As its name indicates, the FT method sets to zero the last third of the Fourier modes ($f(k) = 0$, for $k > 2N/3$), similar to a sharp spectral cut-off filter. On the other hand, the FS method consists on a progressive attenuation of the higher frequencies using the weighting function $f(k) = e^{-36k^{36}}$ (Hou and Li, 2007). Figure 1 presents the spectral function $f(k)$ for the two different methods. Note that both the FT and FS methods behave like a low-pass filter. Although the FT method (continuous line) sets to zero any coefficient larger than $k/k_N > 2/3$, the FS method (dashed line) keeps all the wavelengths unperturbed up until $k/k_N > 3/4$, and then progressively damps the amplitude of the higher wave-number terms. The advantage of both of these methods is that they avoid the need for padding the Fourier partial sums, and hence reduce the numerical cost. Specifically, the computational cost of these methods is $(15/2)N^3 \log_2(N)$ (Canuto



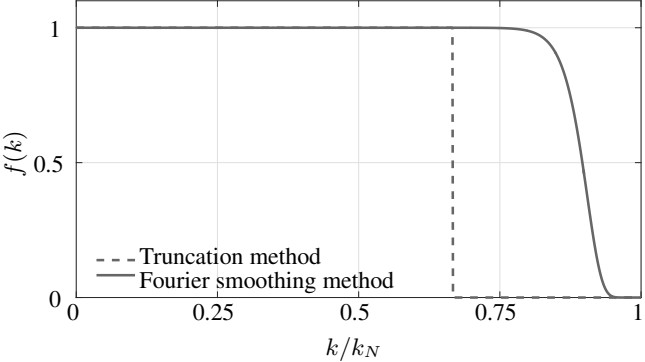

**Figure 1.** Weighting functions used in the FT method (dashed kine) and the FS method (continuous line). The FT method filters scales with $k/k_N > 2/3$ and the FS method at $k/k_N > 3/4$.

et al., 2006), resulting in methods 33% less computationally expensive than the 3/2-rule. The drawback of such approximate approaches is however the fact that a filtering operation is applied to the advection term, resulting in a loss of information. A desirable property of the FS technique when compared to the FT method is that the former exhibits a more localized error and is dynamically very stable (Hou and Li, 2007), while the latter tends to generate oscillations on the whole domain.

## 3 Large-eddy simulation framework

### 3.1 Equations and boundary conditions

The LES approach consists of solving the filtered NS equations, where the time- and space-evolution of the turbulent eddies larger than the grid size are fully resolved, and the effect of the smaller ones is parametrized. Mathematically, this is described through the use of a numerical filter that separates the larger, energy containing eddies from the smaller ones. Often, the numerical grid of size $\Delta$ is implicitly used as a top-hat filter, hence reducing the computational cost (Sagaut (2006), see Moeng and Sullivan (2015) for an overview of the technique in ABL research). As a result, the velocity field can be separated in a resolved component ($\tilde{u}_i$, where $i = 1, 2, 3$) and an unresolved contribution ($u_i'$) (Smagorinsky, 1963). For this technique to be successful, the low-pass filter operation must be performed at a scale smaller than the smallest energy containing scale, deep in the inertial sub-range according to Kolmogorov's hypothesis (Kolmogorov, 1968; Piomelli, 1999). In atmospheric boundary-layer flow simulations this requirement is known to hold in the bulk of the flow, where contributions from the Sub-Grid Scale (SGS) motions model (or sub-filter scale motions) to the overall dissipation rate are modest. In the near surface regions such requirement is not met, as the characteristic scale of energy-containing motions $\mathcal{L}$ scales with the distance from the wall ($\mathcal{L} \approx \kappa z$, where $\kappa \approx 0.4$ is the Von Kármán constant, and z is the wall-normal distance from the wall ), hence remaining an active research field (Sullivan et al., 1994; Meneveau et al., 1996; Porté-Agel et al., 2000; Hultmark et al., 2013; Lu and Porté-Agel, 2014). In this work, the rotational form of the filtered NS equations are integrated, ensuring conservation of mass of the



inertial terms (Kravchenko and Moin, 1997). The corresponding dimensional form of the equation reads as

$$\partial_i \tilde{u}_i = 0, \tag{7}$$

$$\partial_t \tilde{u}_i + \tilde{u}_j \left( \partial_j \tilde{u}_i - \partial_i \tilde{u}_j \right) = -\partial_i \tilde{p*} - \partial_j \tau_{ij}^{\Delta} + \tilde{f}_i. \tag{8}$$

In these equations, $\tilde{u}_i$ are the velocity components in the three coordinate directions $x, y, z$ (stream-wise, span-wise, and vertical
respectively), $\tilde{p}^*$ denotes the perturbed modified pressure field defined as $\tilde{p}^* = \tilde{p} + \frac{1}{3}\rho_0 \tau_{kk}^{\Delta} + \frac{1}{2}\tilde{u}_j \tilde{u}_j$, where the first term is the
kinematic pressure, the second term is the trace of the sub-grid stress tensor and the last term is an extra term coming from the
rotational form of the momentum equation. Here, $\tilde{f}_i$ represents a generic volumetric force. The flow is driven by a constant
pressure gradient in the stream-wise direction imposed through the body force $\tilde{f}_i$. The sub-grid stress tensor is defined as
$\tau_{ij}^{\Delta} = \widetilde{u_i u_j} - \tilde{u}_i \tilde{u}_j$, where the deviatoric components are written using an eddy-viscosity approach

$$\tau_{ij}^{\Delta,d} = \tau_{ij}^{\Delta} - \frac{1}{3}\tau_{kk}^{\Delta}\delta_{ij} = 2\nu_T \tilde{S}_{ij}, \tag{9}$$

with $\nu_T = (C_S \Delta)^2 |\tilde{S}|$ being the so-called eddy-viscosity, $\tilde{S}_{ij} = \frac{1}{2}\left( \partial_j \tilde{u}_i + \partial_i \tilde{u}_j \right)$ the resolved strain rate tensor, and $C_S$ the
Smagorinsky coefficient, a dimensionless proportionality constant (Smagorinsky, 1963; Lilly, 1967). Many studies have in-
vestigated the accuracy of this type of models, showing good behaviour for free-shear flows (Lesieur and Metais, 1996). For
boundary layer flows, the Smagorinsky constant model is over-dissipative close to the wall, since the integral length-scale
scales with the distance to the wall. Therefore, to properly capture the dynamics close to the surface, the Mason-Thompson
damping wall function is used (Mason, 1994). This function is given by $f(z) = \left( C_o^n (\kappa z)^{-n} + \Delta^{-n} \right)^{-\frac{1}{n}}$, and is used to de-
crease the value of $C_S$ close to the wall, reducing the sub-grid dissipation. Note that the molecular viscous term has been
neglected, consistent with the idea of flow over fully rough surfaces, whose drag is parameterized via the inviscid logarithmic
equilibrium law assumption (see below).

The drag from the underlying surface is entirely modeled in this application through the equilibrium logarithmic law for
rough surfaces (Kármán, 1931; Prandtl, 1932), with

$$\tau_W = \left[ \frac{\kappa}{\log(\Delta z / 2 z_0)} \right]^2 \left( \langle \tilde{u}_1 \rangle^2 (\Delta z / 2) + \langle \tilde{u}_2 \rangle^2 (\Delta z / 2) \right). \tag{10}$$

In equation (10), $\langle \tilde{u}_i \rangle$ is the planar averaged velocity sampled at $\Delta z / 2$, $z_0$ is the hydrodynamic roughness length, representative
of the underlying surface, $\Delta z$ denotes the vertical grid stencil, and $\kappa = 0.41$ is the von Kármán constant. The wall shear-stress
is computed considering the module of the horizontal velocity and it is projected over the horizontal directions using the unit
vector $n_i$, such that $\tau_{W,i} = \tau_W n_i$, where

$$n_i = \frac{\langle \tilde{u}_i \rangle (\Delta z / 2)}{\sqrt{\langle \tilde{u}_1 \rangle^2 (\Delta z / 2) + \langle \tilde{u}_2 \rangle^2 (\Delta z / 2)}}, \text{ for } i = 1, 2. \tag{11}$$

In addition, the corresponding vertical derivatives of the horizontal mean velocity field are imposed at the first grid point of the
vertically staggered grid (Albertson et al., 1995; Albertson and Parlange, 1999).

This setup has now been extensively used to study neutrally stratified ABL flows (Cassiani et al., 2008; Brasseur and Wei,
2010; Abkar and Porté-Agel, 2014; Allaerts and Meyers, 2017). Moreover, it is used as foundation to build more complex




simulation of the ABL adding scalar (passive or active) transport (for example, see Saiki et al. (2000), Stoll and Porté-Agel (2008), Calaf et al. (2011), or Salesky et al. (2016)), and many more.

### 3.2 Numerical implementation and time integration scheme

The equations are solved using a pseudo-spectral approach, where the horizontal derivatives are computed using discrete
Fourier transforms and the vertical derivatives are computed using second order-accurate centered finite differences on a stag-
gered grid. A projection fractional-step method is used for time integration following Chorin's method (Chorin, 1967, 1968).
The governing equations become decoupled and the system of partial differential equations can be solved in two steps: at first,
the non-linear advection-diffusion equation is explicitly advanced, and subsequently the Poisson equation is integrated (the
so-called pressure correction step). The latter equation is obtained by taking the divergence of the first equation and setting the
divergence of velocity at the next time step equal to zero, to ensure a divergence free flow field. The algorithm is detailed in the
rest of the section. Initially, the code computes the velocity tensor $\tilde{G}_{ij}^t = \partial_j \tilde{u}_i^t$, which contains all the derivatives of the flow
field required to compute the SGS stress tensor $\tau_{ij}^{\Delta,t} = -2(C_S\Delta)^2|\tilde{S}^t|\tilde{S}_{ij}^t$. In the first step of the projection method, the NS
equations are solved without the pressure. Hence, the intermediary right hand side is computed as

$$\widetilde{RHS}_i^* = \left[ \tilde{u}_j^t \left( \partial_j \tilde{u}_i^t - \partial_i \tilde{u}_j^t \right) - \partial_j \tau_{ij}^{\Delta,t} \right]. \tag{12}$$

Next, an intermediary step is computed using an Adams-Bashforth scheme, following

$$\tilde{u}_i^* = u_i^t + \Delta t \left( \frac{3}{2} \widetilde{RHS}_i^* - \frac{1}{2} \widetilde{RHS}_i^{t-\Delta t} \right), \tag{13}$$

where $\widetilde{RHS}_i^{t-\Delta t}$ is the right hand side of the previous step. At this point, the resulting flow field is not divergence free yet.
The modified pressure is used to impose this fundamental property of the flow filed. Therefore, $\tilde{p}^{*t}$ is computed solving the
Poisson equation

$$\partial_j \partial_j \tilde{p}^{*t} = \partial_k \Gamma_k^t, \tag{14}$$

obtained by taking the divergence of the NS equations. The term $\Gamma_k^t$ in the right hand side of the equation above is given by

$$\Gamma_k^t = \left( \frac{2}{3\Delta t} \right) \tilde{u}_k^t - \frac{1}{3} \widetilde{RHS}_k^{t-\Delta t}. \tag{15}$$

The new flow field for the complete time step is obtained by $\tilde{u}_i^t = \tilde{u}_i^* - \frac{3}{2}\Delta t \partial_i \tilde{p}^{*t}$. Finally, the new right hand side is updated
with the pressure gradient as $\widetilde{RHS}_i^{t+\Delta t} = \widetilde{RHS}_i^* - \partial_i \tilde{p}^{*t}$.

Embedded within this approach, periodic boundary conditions are imposed on the horizontal $(x, y)$ directions. To close the
system, a stress free lid boundary condition is imposed at the top of the domain and an impermeability ($\tilde{w} = 0$) condition is
imposed at the lower boundary, which sums to the parametrized stress described in section 3.

For time-integration performance, the code is parallelized following a 2D pencil decomposition paradigm similar to the one
presented in Sullivan and Patton (2011), partitioning the domain into squared cylinders aligned along one of the horizontal





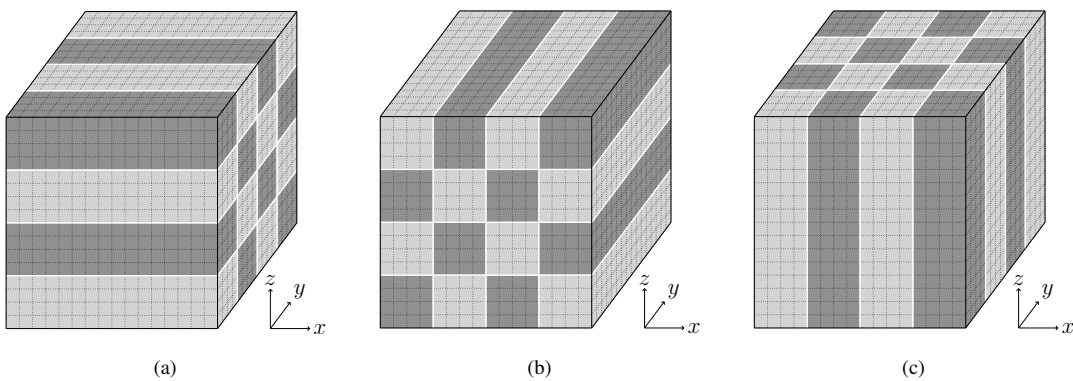

**Figure 2.** 2D Pencil decomposition of the computational domain with the domain transposed into the 3 direction of space: (a) X-pencil, (b) Y-pencil, (c) Z-pencil (inspired form (Li and Laizet, 2010)).

directions, as shown in figure 2. This parallelization scheme allows bigger computational domains when compared to the layer-based 'z-slice' decomposition schemes (Kumar et al., 2006; Kumar, 2007), where the domain was only split in horizontal slices. The 2D stencil decomposition is implemented using the 2DECOMP & FFT open source library (Li and Laizet, 2010), which shows exceptional scalability up to a large number of MPI processes (Margairaz et al., 2017).

**3.3   Analysis of the numerical cost**

The LES algorithm can be separated into four distinct modules: (1) computation of the velocity gradients, (2) evaluation of the SGS stresses and (3) of the convective term, and (4) computation of the pressure field by solving the Poisson equation. These four modules represent the bulk of the computational cost of the code, in addition to MPI communication. Figure 3 presents a simplified flowchart of the main algorithm with each of the four modules.

The four modules have been individually timed to evaluate their corresponding computational cost at a resolution of $N_x \times N_y \times N_z = 128^3$. Results are shown in figure 4. As it can be observed, more than half of the integration time step ($\sim 60\%$) is spent computing the convective term. The three other modules share the rest of the integration time as follows: the computation of the velocity gradients ($\sim 20\%$), the Poisson solver ($\sim 16\%$) and the SGS ($\sim 4\%$). It is important to note that this test was conducted without any I/O as it is not relevant to assess the computational cost of the momentum integration. As explained

in section 2, the non-linear term requires the use of dealiasing techniques to control the aliasing error, which traditionally are associated with a padding operation (as mentioned in Section 2), and hence higher computational cost. It is important to note that although the overall integration time distribution between each individual module might vary depending on the numerical resolution employed (*i.e.* the cost of the Poisson solver is expected to increase with resolution increases), the overall cost of the convective term will remain important regardless of the changes in numerical resolution. The goal of this work is to explore the

possibility of using alternative dealiasing techniques to enhance the computational performance, while maintaining accurate turbulent flow statistics in simulations of ABL flows. It is important to note that the SGS model used here takes a relatively





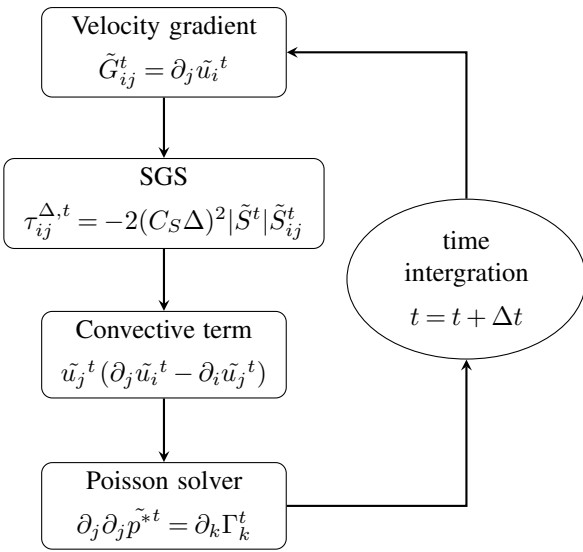

**Figure 3.** Simplified flowchart of the main algorithm presenting the four modules that represent the bulk of the computational cost.

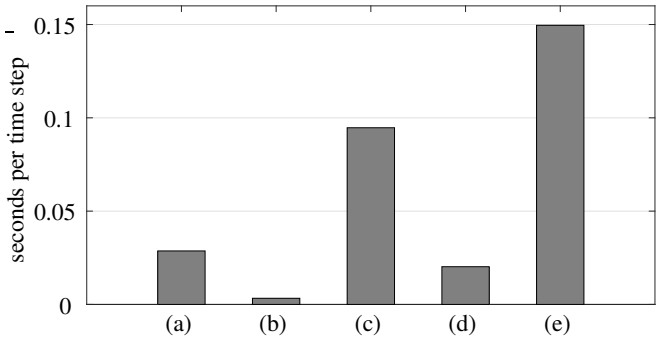

**Figure 4.** Individual timing of the 4 modules of the time loop averaged over 10k steps: (a) velocity gradient, (b) SGS, (c) convective term, (d) Poisson solver, and (e) total time loop. The numerical resolution is $128^3$, run with 64 MPI processes, and a domain decomposition of $8 \times 8$

small fraction of the time integration. This fraction is likely to be larger if a more sophisticated model is used, for example the dynamic Smagorinsky model (Germano et al., 1991) or the Lagrangian scale dependant model (Bou-Zeid et al., 2005).

### 3.4 Study cases

The goal of this study is to develop a cost benefit analysis for the different, already established, dealiasing methods from a
5   computational cost stand point as well as in terms of accuracy in reproducing turbulent flow characteristics. For this reason, three different cases have been considered corresponding to the three dealiasing methods considered: (a) the 3/2-rule used as reference, (b) the Fourier truncation method (FT), and (c) the Fourier smoothing method (FS). All the simulations have been run with a numerical resolution of $N_x \times N_y \times N_z = 64^3$, $128^3$, and $256^3$ with a domain size of $(L_x, L_y, L_z) = (2\pi, 2\pi, 1) \cdot z_i$,





where $z_i$ is the height of the boundary layer taken here with a value of $z_i = 1000$ m. A uniform surface roughness of value $z_0/z_i = 10^{-5}$ is imposed, which is representative of sparse forest or farmland with many hedges (Stull, 1988) (p.380). The simulations have been initialized with a vertical logarithmic profile with added random noise for the $\tilde{u}_1$ component. The two other velocity components $\tilde{u}_2$ and $\tilde{u}_3$ have been initialized with an averaged zero velocity profile with added noise to generate

the initial turbulence. The integration time step is set to $\Delta t = 0.2$ s for both $64^3$, and $128^3$ simulations and to $\Delta t = 0.1$ for the $256^3$ simulation. These time steps have been set to keep the equivalent CFL number bounded throughout the simulation. The Smagorinsky constant and the wall damping exponent are set to $C_S = 0.1$ and $n = 2$ (Mason, 1994; Porté-Agel et al., 2000; Sagaut, 2006).

For each dealiasing method, the simulations at $64^3$, and $128^3$ were run until the flow reached statistic convergence of the
friction velocity $u_*$ and the mean kinetic energy. This warm-up time was fixed to $\sim 95\,T$ (where $T$ is the flow-through time, defined as $T = U_\infty\,t/L_x$). At this point, running averages were computed to evaluate the different flow statistics presented in the following sections. The simulations were run for an additional $\sim 190\,T$, providing long enough averaging times. The $256^3$ simulation was run for $\sim 67\,T$ because of computational resource restrictions. In parallel, runs with higher horizontal resolution were used to evaluate the computational cost of the dealiasing methods with increasing numerical resolution (timing
runs). These last simulations were only run for a few thousand iterations. Table 1 contains a summary of all the simulations preformed in this work.

| Simulation type | Resolution $N_x \times N_y \times N_z$ | Flow-through time $T$ |
|---|---|---|
| Statistics runs | $64 \times 64 \times 64$ | $\sim 285\,T$ |
| | $128 \times 128 \times 128$ | $\sim 285\,T$ |
| | $256 \times 256 \times 256$ | $\sim 67\,T$ |
| Timing runs | $128 \times 128 \times 128$ | $\sim 2\,T$ |
| | $256 \times 256 \times 128$ | $\sim 2\,T$ |
| | $512 \times 512 \times 128$ | $\sim 2\,T$ |
| | $1024 \times 1024 \times 128$ | $\sim 2\,T$ |
| | $2048 \times 2048 \times 128$ | $\sim 2\,T$ |

**Table 1.** Simulations summary, each simulation was run with the three different dealiasing methods.

## 4    Results

### 4.1    Computational cost

To evaluate the effectiveness of the FT and FS methods, the computational cost of these two methods has been computed and
compared to the computational cost of the original 3/2-rule method. Figure 5 presents the corresponding cost of the FT and





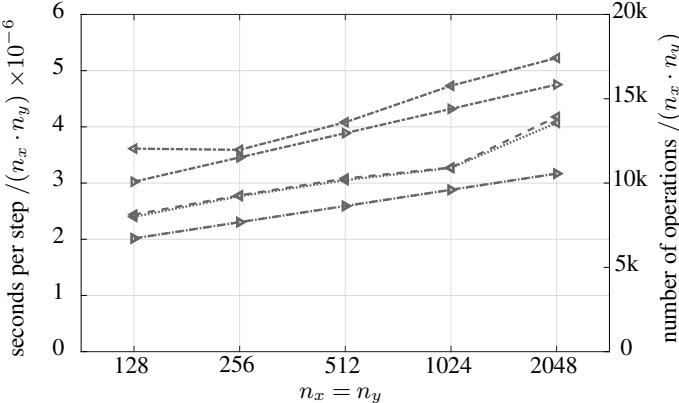

**Figure 5.** Computational cost of the convective module as a function of the horizontal resolution. The timing of the module is presented on the left vertical axis and represented by left-pointing arrows. The number of operation is shown on the right vertical axis and represented by right-pointing arrows. The three different dealiasing methods are plotted as 3/2-rule in dot-dashed line, FT method in dotted line and FS method in dashed line. The numerical resolutions are $n_x \times n_y \times 128$, run with 64 MPI processes, and a domain decomposition of $8 \times 8$

FS methods for the convective term as it is the only one affected by the dealiasing method. The horizontal resolution has been increased from $128 \times 128$ to $2048 \times 2048$ to evaluate the performance across different grid resolutions. Only the horizontal resolution is changed given that the vertical direction is treated in physical space with finite difference methods and does not require any dealiasing treatment because the truncation error associated with the second order centered finite difference

method decreases the aliasing errors (Kravchenko et al., 1996; Canuto et al., 2006). In figure 5, the ordinate axis are divided by $n_x \cdot n_y$ to show the effect of the increase in resolution on the computational cost. The number of MPI processes and the domain decomposition have been kept identical to avoid introducing effects from the parallelization scaling into the results. Hence, only the effect of the resolution change on the computation time of the dealiasing methods is presented here. Results confirm that the computational cost of the convective term is significantly smaller when using the FT and FS dealiasing methods, with

gains between 20% and 30%, depending on the numerical resolution. The results follow the predicted computational cost predicted by the number of operations presented in section 2. The small deviation in the computational cost present in figure 5 is the result of the varying load of the computer cluster since all simulations were run using the same number of nodes to avoid having to add the code's scaling properties to the analysis. From the results it is also important to note that there is no significant difference in the computational cost between the FT and FS methods given that both use the same grid size and

hence the corresponding numerical complexity of both methods is similar. Therefore, as initially expected, results show that the less used FT and FS dealiasing methods have a clear impact on the performance of the code, yielding a net gain between 20% and 30% over the traditional 3/2-rule. It is also important to note that these methods are quite simple to implement given that there is no need for artificially extending the numerical grid.



In the following subsections, we compare the effects of the different dealiasing schemes on the turbulent flow characteristics and quantify the differences. The goal is to clarify the extend of the perturbations introduced on the flow field if the more economical and less cumbersome FT and FS dealiasing techniques were to be implemented.

### 4.2 Flow statistics

Traditional flow metrics are investigated next and compared between the different dealiasing schemes. Apart from figure 6 which shows the $256^3$-case, results for the $128^3$-case are presented in this section. The findings of the $64^3$-case are not shown because they are very similar to the $128^3$-case. The results are normalized using the traditional scaling variables: the friction velocity ($u_*$) and the boundary layer height ($z_i$). As a starting point, figure 6 shows an instantaneous snapshot (pseudo-color plots) of the strea-mwise velocity perturbation for the three dealiasing methods. An additional case without dealiasing in the convective term was run (not showed here). In the latter case, the flow completely laminarized throughout, destroying the turbulence, highlighting the extreme importance of the dealiasing operation (Kravchenko and Moin, 1997). On the contrary, when dealiasing schemes are applied, the instantaneous flow field appears quite similar among the different cases. Irrespective of the dealiasing method that is used, stream-wise elongated uniform high- and low-momentum bulges flank each other, as apparent in figure 6. This is a common phenomena in pressure-driven boundary-layer flows (Munters et al., 2016). Qualitatively, small differences can be appreciated on the structure and distribution of the smaller-scale turbulence within the flow. This is expected given the explicit filtering used in the latter dealiasing schemes (the FT and FS). For example, the flow in subplot *b* is showing the effect of the cut-off filter, as small circular structures can be observed. Thus, the FT methods is changing the nature of the small scales in the turbulent flow field. Alternatively, the stream-wise velocity obtained with the FS method does not seem to be as affected by these artifacts. Note however that these differences are in the instantaneous field, whose intrinsic nature is unsteady and chaotic. A more quantitative analysis follows.

A step in that direction is made through the analysis of the averaged velocity profiles and the velocity gradients, shown in figure 7. Profiles show a good logarithmic behavior within the surface-layer ($z \sim 0.1z_i$) for all cases, matching the imposed boundary conditions, and represented with the solid line. Specifically, the FT method exhibits a good overlap up to $z \sim 0.4z_i$ with the velocity profile obtained using the 3/2-rule. A maximum momentum departure of $\sim 14\%$ is measured at the top of the domain. On the other hand, the FS method presents a weaker differentiated behavior of only $\sim 7\%$ from $z \sim 0.05z_i$ up to the top of the domain. These overestimation of the mean stream-wise velocity above the surface layer by the FS method leads to an overall increase of $\sim 15\%$ on the mean kinetic energy (MKE) of the system. Otherwise, when using the FT method the overshoot in mean velocity seems to be reduced to the top of the numerical domain, hence leading to a weaker overall increase in MKE of $\sim 11\%$. This increase in MKE can be related to the low-pass filtering operation that is performed in the near-wall regions by the FT and FS methods, which modify the balance between turbulent and laminar stresses at such location, in favor of the laminar component, thus resulting in a higher mass flux. Note that the low-pass filtering operation performed by the FT and FS methods has similar effects to increasing the LES eddy-diffusivity.

Complementary to the mean velocity profiles, figure 7 (b) presents the normalized velocity gradient, $\Phi_m = \kappa \frac{z}{u_*} \partial_z \langle U \rangle_{xy}(z)$. In all cases, it presents a large overshoot near the surface, which is a well known LES problem of wall bounded flows and has

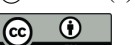



**Figure 6.** Instantaneous streamwise velocity perturbation $u'(x, y, z, t) = u(x, y, z, t) - \bar{u}(x, y, z)$ at $z/z_i = 0.027$ for the three different dealiasing methods: (a) 3/2-rule, (b) FT method, and (c) FS method. The numerical resolutions is $256^3$.



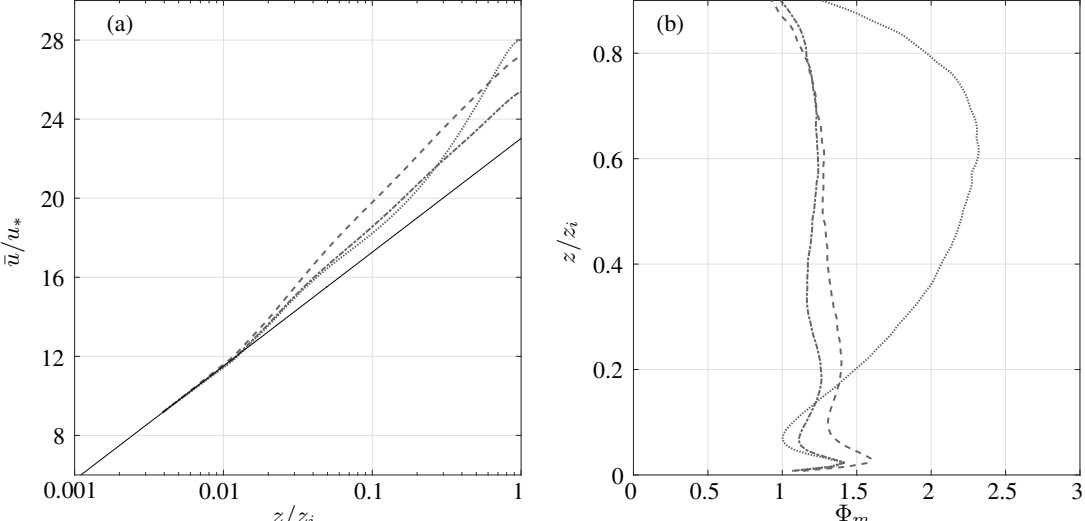

**Figure 7.** Plots of the non-dimensional mean streamwise velocity profile (a) and the mean stream-wise velocity gradient (b) for the three different dealiasing methods: 3/2-rule in dot-dashed line, FT method in dotted line and FS method in dashed line. The solid line in (a) represents the ideal log-law profile.

been extensively discussed in the literature (Bou-Zeid et al., 2005; Brasseur and Wei, 2010; Lu and Porté-Agel, 2013). The high gradient near the wall suggests that the resolved kinetic energy is over dissipated by the Smagorinsky model close to solid boundaries, even with the Mason-Thompson damping wall function. Therefore, the overall turbulent stresses are too low there, resulting in an overestimated mean velocity in the bulk of the flow. Again, such over-dissipative behavior is further aggravated

when applying the FS or the FT methods. From the vertical gradient profiles it can be observed that the three methods produce similar results within the surface layer. Above that, the FS follows very well the profile provided by the 3/2-rule. Contrary, the truncation method over-estimates the velocity gradient starting from $z = 0.2z_i$ up to the top of the domain. This is an important result because it shows that despite the over-dissipative behavior of the FS method, its vertical distribution remains similar to that obtained with the 3/2-rule. Contrary, with the FT method there exists both, a stronger over-dissipation of turbulent motions

in the near wall regions, inducing a higher mass flux, and a differentiated momentum distribution throughout the domain. This results from the fact that when using the FT method a broader range of energetic near-wall coherent structures are removed, hence having a larger effect on the overall vertical distribution of momentum.

Next we investigate the effect of the simpler and computationally faster dealiasing-schemes on the second order moments. The vertical structure of variances $\overline{u'u'}, \overline{v'v'}, \overline{w'w'}$ and of the turbulent stress $\overline{u'w'}$ are featured in figure 8, where $\overline{(\cdot)}$ denotes the

space + time averaging operator, and $(\cdot)'$ denotes a fluctuation from the corresponding average value. Note that when averaging in space and (subsequently) in time, dispersive terms are embedded in the variance and covariance terms (Raupach et al., 1991; Finnigan, 2000). The resulting profiles are comparable to those found in previously published LES studies (Porté-Agel et al., 2000; Bou-Zeid et al., 2005).





**Figure 8.** Profiles of the non-dimensional variances (a/b/c) and shear stress (d) for the three different dealiasing methods: 3/2-rule in dot-dashed line, FT method in dotted line and FS method in dashed line.

Interestingly, results of the vertical flux (or stress, resolved and SGS) of stream-wise momentum (figure 8(d)) illustrate a good agreement between the different scenarios. Contrary, larger differences appear in the diagonal components of the turbulent stress tensor. For example, the stream-wise variance is underestimated by the FT method in the surface-layer and up to almost $0.5z_i$. This underestimation is probably the one causing the apparent differences earlier observed qualitatively in the instantaneous velocity fields from Figure 6. On the other hand, the cross-stream and vertical variances are both slightly overestimated by the FS and FT methods. In all cases, the overestimation ceases near $0.5z_i$, and while for the cross-stream variances





initiates at the surface, the overestimation on the vertical variances initiates towards the top of the surface-layer. It is important to note again, that the FS method, while slightly diverging in absolute values (maximum divergence of 15%), it presents a very similar vertical distribution of the variances compared to the reference values obtained with the 3/2 rule. Contrary, the FT method diverges both, in absolute magnitude (maximum divergence of 35%) and vertical distribution, specially in the vertical

variance. Once again, this is the result of the different filtering operations applied to the small turbulence scales during the integration of the NS equations.

To complement the analysis of the effect of the different dealiasing methods on the physical structure of the flow, the corresponding power spectra is investigated. According to Kolmogorov's energy cascade theory, the inertial sub-range of the power spectra should be characterized by a power law of −5/3 slope (Kolmogorov, 1968). In this range the effects of

viscosity, boundary conditions, and large scale structures are not important. Also, in wall-bounded flows with neutral buoyancy, a production range should also be present, following a power-law scaling of −1 (Gioia et al., 2010; Katul et al., 2012; Calaf et al., 2013). Figure 9 shows the energy spectra of the stream-wise velocity obtained using the different dealiasing methods. The spectrum obtained using the 3/2-rule matches well the traditional turbulent spectra presented in the literature (Cerutti, 2000; Bou-Zeid et al., 2005) and it is used to assess the effects introduced by the FT and FS dealiasing methods. From this spectral

analysis, it can be observed that low wave-number modes in the spectra are very similar between the 3 methods and only the high wave-number ranges are modified. The dealiasing methods have been designed for such behavior, since only the higher frequencies are filtered. From the FT method flow field spectra it is clearly visible the effect of the cut-off applied within the dealiasing scheme. It is apparent that the capacity of the LES solver to reproduce the fine scale turbulence structure of the flow is strongly jeopardized when using the FT method and limited at the scale of $3/2 \cdot \Delta$ close to the LES filter scale $\Delta$. Essentially,

this method artificially over-dissipates the turbulent kinetic energy and yields to an overestimation of the mean kinetic energy. In contrast, and as one would expect, the energy spectrum obtained using the FS method does not produce such a large energy cut-off because the high wave-numbers are only smoothed to attenuate the effects of the aliasing errors. Therefore, a larger range of the spectrum is resolved and less turbulent kinetic energy is dissipated by aliasing errors. In conclusion, the FS method produces an energy spectra that reproduces most of the observed features of the fully dealiased spectra (3/2-rule) at a reduced

computational cost.

## 5  Conclusions

The Fourier-based pseudo-spectral collocation method (Orszag, 1970; Orszag and Pao, 1975; Canuto et al., 2006) remains the preferred "work-horse" in simulations of wall-bounded flows over horizontally periodic regular domains, which is often used in conjunction with a centered finite-difference or Chebychev polynomial expansions in the vertical direction (Shah and

Bou-Zeid, 2014; Moeng and Sullivan, 2015). This approach is often used because of the high-order accuracy and the intrinsic efficiency of the fast-Fourier-transform algorithm (Cooley and Tukey, 1965; Frigo and Johnson, 2005). In this technique, the leading-order error term is the aliasing that arises when evaluating the quadratic non-linear term in the NS equations. Aliasing errors can severely deteriorate the quality of the solution, even leading to laminarization of turbulent flows, and hence need to be





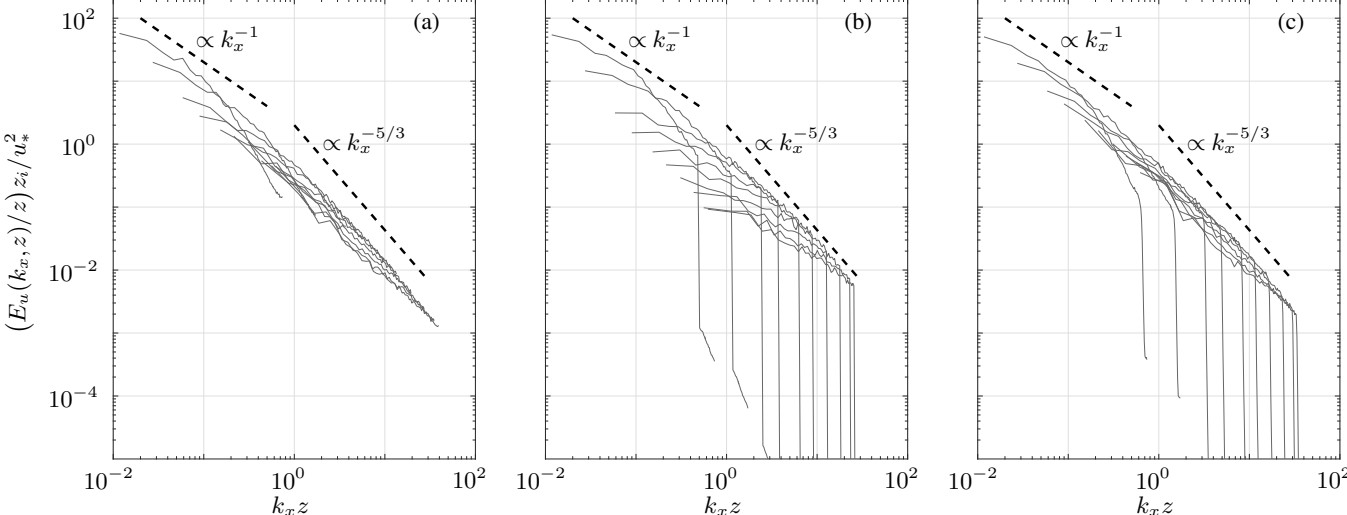

**Figure 9.** Normalized stream-wise spectra of the stream-wise velocity as a function of $k_x z$ for the three different dealiasing methods: 3/2-rule (a), FT method (b), FS method (c) at height $z/z_i = 0.0117, 0.0273, 0.0586, 0.0898, 0.1523, 0.2148, 0.3086, 0.4336, 0.5586$, and $0.6211$.

treated adequately. In this work a performance/cost analysis has been developed for three well-accepted dealiasing techniques (3/2-rule, FT and FS) to evaluate the corresponding advantages and limitations. Note that while the FT method applies a cut-off filter at the scale of $3/2 \cdot \Delta$ (close to the LES filter scale $\Delta$), the FS method uses a smooth filter to attenuate the effects of the aliasing errors, hence progressively attenuating the effect of the smaller turbulent scales.

A very important result is the fact that computational cost is reduced by 20% to 30% when using the FS or FT methods, respectively, depending on the numerical resolution. In this regard, results are aligned with the theoretical predictions despite of parallelization and domain decomposition, indirectly illustrating the robustness of the LES platform here used.

To evaluate the influence of these two dealiasing schemes beyond the computational cost, traditional flow statistics have been evaluated. The results illustrate that the FT method over-dissipates the turbulent motions in the near wall regions, inducing a

higher mass flux, and a different velocity distribution throughout the domain. Contrary, when comparing the FS method against full dealiasing, similar vertical distributions in momentum and second-order statistics are obtained, despite the additional dissipation that such approximate methods introduce (via damping of small scale motions). These differences in flow statistics between the FT and FS methods are the result of the sharp low-pass filter applied in the FT method compared to the smoothing function used in the FS method. Although the simulations presented in this paper do not account for the Coriolis Effect (flow

rotation) results are very important for the ABL-modeling community. The effects of the dealiasing schemes affect more the small scale turbulence, and are mostly encountered in the surface layer, where rotational effects are minimal.

The results presented here show compelling evidence of the benefits of the FS method, providing important computational gains while producing similar instantaneous turbulence behavior (spectral analysis) and converged statistics to the commonly accepted 3/2-rule (full dealiasing).





*Code availability.* The sources of the LES code developed at the University of Utah are accessible in pre-release at https://doi.org/10.5281/zenodo.1048338 (Margairaz et al., 2017).

*Data availability.* Due to the large amount of data generated during this study, no lasting structure can be permanently supported where to openly access the data. However, access can be provided using the Temporary Guest Transfer Service of the Center of High Performance

Computing of the University of Utah. To get access to the data, Marc Calaf (marc.calaf@utah.edu) will provide temporary login information for the sftp server.

For reviews:

| | |
|---|---|
| sftp server: | `guest-transfer.chpc.utah.edu:/scratch/margairaz-gmd-2017-272` |
| username: | `gtx0020` |
| password: | `byoG7jP1g` |

*Competing interests.* The authors declare no competing interests

*Acknowledgements.* F.M. and M.C. acknowledges the Mechanical Engineering Department at University of Utah for start-up funds. M.G.G and M.B.P. acknowledge the Swiss National Science Foundation (SNSF-200021- 134892), and the Competence Center for Environmental Sustainability (CCES-SwissEx) of the ETH domain. The authors would also like to recognize the computational support provided by the Center for High Performance Computing (CHPC) at University of Utah as well as the Extreme Science and Engineering Discovery Environment (XSEDE) platform (project TG-ATM170018).





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
