# Peer review of "Comparison of dealiasing schemes in large-eddy simulation of neutrally-stratified atmospheric flows"

_Geoscientific Model Development, 2017_

## Referee Comment (RC1) · Anonymous Referee #1 · 8 Feb 2018

General comments: This manuscript addresses the computational cost and the consequences of using different dealiasing schemes in the advective term of the momentum equations for pseudo-spectral discretizations, such as those frequently used in simulations of turbulence in the planetary boundary layer. In particular, two more economical approaches are contrasted with the exact 3/2 rule. Overall the results presented may be relevant for the community interested in turbulence simulation in geophysical settings, such as the planetary boundary layer and the ocean mixed layer. The paper is well written, even though a more concise version would probably be more appealing to the reader. I am a bit surprised by the low cost of the Poisson solver, and the choice of a very low-cost subgrid-scale model certainly helps increasing the share of

the convective term on the total cost (as briefly mentioned in the text). After reading the manuscript, I would probably still stick to the 3/2 rule, but that is just my opinion. Having a lower-cost alternative may be useful to other scientists working in the field. The topic may be a bit on the margins of interest of GMD, but I do recommend it for publication. Some important remarks follow below.

Specific comments:

1. The authors perspective on the FT method is a bit different from mine, and to be honest, I am not sure which one is the most prevalent in the community. I have always considered the FT method (which is also known as the 2/3 rule) as a slightly different implementation of the exact 3/2 rule, but with a reduced number of effective grid points (actually Neff=2N/3 instead of N). In my view, performing a calculation with 3N/2 points using the FT method should be identical (in simulation results but maybe not in computational cost) to a simulation with N points using the 3/2 rule, no? I also think the that simulations using the FT method should actually report grid resolution based on Neff and not on N (i.e. the grid spacing should be Dxeff=3Dx/2 instead of Dx), but that is usually not done. Maybe the authors can comment on this?

2. I am surprised by the high share of the cost carried by the convective term compared to a very low share carried by the Poisson solver. Is this something that is specific to the pencil decomposition parallelization? It may be the case that padding with the 3/2 rule is not as simple in a pencil decomposition approach given that one usually needs to pad the entire 2D "horizontal" wavenumber space? Maybe some details of the padding in the context of pencil decomposition would be useful. In addition, is it possible that the Poisson solver is faster when pencil decomposition is used?

3. The quantification of computation cost is section 4.1 is fine. Regarding the results in section 4.2, I wonder if the question could be posed the other way around. In my view, maybe the most relevant question would be: "given a set computational cost, do I get better results (a) using the 3/2 rule or (b) running a finer grid using a less

expensive dealiasing?". I know the experiment is not easy to design (because one needs to estimate the computational cost ahead of time), but that seems to be the more important question to be answered. If I save 30% on dealiasing and spend it on more resolution, do I get better results? From my comment above, I would expect that the FT method is almost equivalent to the 3/2 rule (if smartly coded). How about the FS?

4. Regarding the interpretation of Figure 7. I am not convinced the log-law prediction is as good as described in the text (but perhaps I am missing something here?). First, it seems that there is one log-law on top of the solid line that may extend only for 2 or 3 vertical levels (ending around $z/z_i <= 0.02$). Then there is a second log-law (only in 3/2 and FT cases) that starts around $z/z_i = 0.04$ and goes beyond $z/z_i = 0.1$, This second log-law has the incorrect roughness (if one were to extrapolate it to $u=0$, it would yield a lower value of $z_0$ than the one imposed on the simulation, I think). There is no clear second log-law in the FS method. I would see this as a concern for the FS method (which is being advocated here), except that not even the exact 3/2 rule has a good log-law (as seen in Bou-Zeid et al, 2005). I am pretty sure this is due to the SGS model adopted here. In any case, if I had to choose between the FS and FT methods based on Figure 7, I would probably go with FT, since it does a reasonable job in the lower 20% of the domain (which is the region of interest in a simulation like this, I guess). Also, the FS method seems more over-dissipative near the wall (on panel b), which is opposite to what is described in the text?

5. Regarding Figure 9. I do not agree that the filtering only affects the small-scale end of the spectra. For the FT case, it seems clear to me that there is significant damping of the large scales as well. This is related to the underestimation in the variance of the streamwise velocity seen in Figure 8. This is worrisome, and probably related to the fact that the true resolution here (Neff=85 points) is too coarse to model the ABL. I wonder if this situation would persist in the 256^3 simulations?

6. I see two facts that could benefit from more discussion in the manuscript (maybe in

the conclusions):

a. The importance of reducing computational cost in the advection term for simulations that use more sophisticated SGS models (maybe this can be brought up again in the conclusions?)

b. The consequence for including one or many additional scalar fields (temperature, water vapor, etc.). These also require dealising and do not require additional pressure solvers and/or very expensive SGS models. I would probably anticipate that in simulations with several scalars, the savings would be significantly larger. Is that correct?

Technical corrections:

1. Page 4, Line 33 - please check that the N^3 term in the cost is correct here.

2. Page 5, Line 10 - delta implicit does not really correspond to a top-hat filter. The properties of an implicit filter are tied to the discretization scheme. As an example, for a true 3D spectral code the implicit filter is a spectral cutoff filter.

---

## Referee Comment (RC2) · E. Bou-Zeid (Referee) · 27 Feb 2018

The paper compares various approaches for the dealiasing of the non-linear terms in large eddy simulation of atmospheric flows. Given that spectral methods are widely used in studying such flows under idealized conditions since they offer higher speed and better accuracy, the general theme is of interest to GMD readers. The paper does a good job in presenting the fundamentals of the problem, the proposed solutions, and how they compare when implemented in an actual code. But major revisions are needed. In particular, the study is a valuable comparison of the methods that is not available (to the best of my knowledge) in the literature and the authors should not try to conclude that one is more optimal than the others. They can simply present their findings and let the users determine which method is suitable for their needs.

**Major comments:**

1. While the FS method seem to be giving an acceptable performance as the authors argue, I wonder whether the ABL LES community should be going in a direction of saving computing time rather than maximizing the accuracy of the computation. We push for higher resolution to gain better accuracy and, with increasing computing power, I wonder whether a 20 % drop in simulation time is worth it. We use dynamic SGS models that increase the computing time by 20% all the time. The plots in Fig 7 do not indicate that the FS method is as good as the 3/2 method. So in general I think the authors should not focus on the conclusion that the FS method is a good surrogate. They should present the information and findings, which will help modelers decide on the tradeoffs they want (on my end this convinces me that using a 3/2 method is indeed worth it.).

2. How do the FT and FS method influence the potential use of dynamic models that require good accuracy on the smallest resolved scales? If as the spectra show they damp these scales, than that would preclude using dynamic models and would be a significant disadvantage of FT and FS. The authors have in their code some dynamic models, they could perform the dynamic computations while still using the Static Smagorinsky (compute a dynamic Cs but don't use it).

3. Fig 8d and the associate sentence "Interestingly, results of the vertical flux (or stress, resolved and SGS) of stream-wise momentum (figure 8(d)) illustrate a good agreement between the different scenarios." The authors should be careful in this interpretation. The constant pressure gradient forcing requires and forces the stress profile to be linear. Regardless of how turbulence ends up looking like the turbulent fluxes have to adjust to balance the mean  $\partial P/\partial x$ . What this figure indicates is that the SGS fraction is not strongly affected by the choice of dealiasing method, which is a good thing.

4. Figure 9, and more generally: I would have liked to see a direct comparison of the largest scales (by filtering all simulations at  $n\Delta$ , where n correspond to start of the damping or cutoff in figure 1) to see if the differences are only on the smallest scales or not (although given the mean velocity profiles, I suspect they are not).

Minor comments

1. Title is long and too descriptive: how about replacing the wordy "atmospheric boundarylayer type flows" with "atmospheric flows". One in fact could foresee using such methods for cloud resolving LES outside the ABL.

Same on last line of abstract: why restrict the applications only to ABL flows?

- 2. Abstract line 3: better to replace "integrating" by "time advancing"
- 3. Abstract lines 4-5: not sure what is meant by "This is of special relevance when using high order schemes." Spectral schemes are always "high order"
- 4. First 3 lines (19-21) of introduction. In fact the # of grid points in LES has not been following Moore's law. See <a href="https://doi.org/10.1017/jfm.2014.616">https://doi.org/10.1017/jfm.2014.616</a> . This shows that the LES community has not been taking full advantage of increasing computing power to improve model accuracy, which I think is a remiss.
- 5. Page 2, line 2. Add comma after "With increasing computer power". I think there are a few other missing commas after introductory phrases.
- 6. Correct **wall bounded flow** to **wall-bounded flow** on page 2 line 7.
- 7. Page 2, Lines 30-32: authors talk about the need to expand the grid onto 3/2N and then say "As a result, due to the non-linear dependence on N" This example make it sound linear. I think they are referring to the non-linearity of the FFT cost with N, which they explain later. Clarify.
- 8. Also maybe they should clarify that if FFTs are used in 2D, the cost rises even more quickly as N rises.
- 9. Page 3 line "via a set of LES of fully developed ABL type flows and with a corresponding comparison on the effect in turbulent flow statistics and topology". Convoluted phrase. Simplify.
- 10. Page 3, line 8 "environmental fluids"
- 11. Page 3, line 19 "thus discouraging its use in most practical situations" are the authors certain of this statement? The 3/2 rule is use very very widely. If not maybe their paper would be a good reference for modelers to see the advatanges of using it.
- 12. Page 5 line 13, and maybe other places. Referring to the production range as the energy containing range is inaccurate and misleading. The statement "For this technique to be successful, the low-pass filter operation must be performed at a scale smaller than the smallest energy containing scale, deep in the inertial sub-range according to Kolmogorov's hypothesis (Kolmogorov, 1968; Piomelli, 1999)." For example makes no sense if that jargon is used. Please fix and use energy production range instead.
- 13. Eq 8, the  $\tau$  should have a d superscript if the trace is already in p\* as the authors write.
- 14. Page 6, line 18: the log law is not inviscid since it is derived from matching the viscous sublayer and the outer layer. If the wall is smooth for example z0 depends on viscosity.
- 15. Page 6 line 25: what does "module" mean? Do they mean modulus?
- 16. Should equation 12 include fi to be consistent ?

- 17. Page 8, lines 10-15: Authors should clarify this is with the baseline 3/2 dealiasing I presume. Also how does the parallelization method impact these numbers?
- 18. Page 10, lines 1-2: The  $z_0$  they impose is 1cm, which corresponds more to a grass field than to a sparse forest of to a farmland. I suggest they check Brutsaert's books rather than to Stull for  $z_0$ .
- 19. Figure 5 and other are difficult to read. Why not use colors for the online version (Color is free with EGU, no?)
- Page 11, lines 9-11: 30% drop in the convective term cost is good but I would not say it is significant. It would only be equivalent to about 20% drop in total computing time (given Fig 1), which would only be equivalent to a 5% reduction in the resolution. So I would remove "significantly" on line 9.
- 21. Page 11, lines 10-11: "the predicted computational cost predicted by" remove one of the "predicted".
- 22. Page12, line 2: replace "extend" with "extent"
- 23. Page12, line 9: correct the misspelling of "stream-wise"
- 24. Page 12 line 25, and page 14 line 10: "differentiated" is an unclear word. Please remove and clarify the two sentences.
- 25. Page 14 lines 15-16. There won't be any dispersive stresses in their simulations over homogeneous terrain so why mention them?

Elie Bou-Zeid

---

## Author Comment (AC1) · 2 May 2018

**Title:** *Comparison of dealiasing schemes in large-eddy simulation of neutrally-stratified atmospheric boundary-layer type flows*

**Authors:** *Fabien Margairaz, Marco G. Giometto, Marc B. Parlange, and Marc Calaf*

**Submitted to:** Geosci. Model Dev. (gmd-2017-272)

**Reviewer:** Anonymous Referee #1

**Date:** May 1, 2018
* * *
**General response**

**Reviewer general comment:** *This manuscript addresses the computational cost and the consequences of using different dealiasing schemes in the advective term of the momentum equations for pseudo-spectral discretizations, such as those frequently used in simulations of turbulence in the planetary boundary layer. In particular, two more economical approaches are contrasted with the exact 3/2 rule. Overall the results presented may be relevant for the community interested in turbulence simulation in geophysical settings, such as the planetary boundary layer and the ocean mixed layer. The paper is well written, even though a more concise version would probably be more appealing to the reader. I am a bit surprised by the low cost of the Poisson solver, and the choice of a very low-cost subgrid-scale model certainly helps increasing the share of the convective term on the total cost (as briefly mentioned in the text). After reading the manuscript, I would probably still stick to the 3/2 rule, but that is just my opinion. Having a lower-cost alternative may be useful to other scientists working in the field. The topic may be a bit on the margins of interest of GMD, but I do recommend it for publication. Some important remarks follow below.*

**Authors response:** We thank the reviewer for his/her positive comments and the recommendation for publication. In this regard the reviewers' comments have made us realize that there were some aspects in the manuscript that required additional clarification and/or improvement. For this reason, down below we provide a clear response to each one of the issues and comments indicated by the reviewer, and an additional explanation on how these have now been addressed within the manuscript.

Based on the comments from both reviewers, we have added a new discussion section to improve the readability of the manuscript. We have also improved the results section by running additional simulations at the resolutions of $192, 192, 128$ and $192^3$, as well as running the $256^3$-case long enough to be able to report converged statistics. As a result Figures 7, 8 and 9 have been updated, and a new figure (Figure 10) has been included, which is discussed at the end of the results section.

**Changes in manuscript:** The new discussion section is reported here:

"In the development of this manuscript, focus has been directed to the study of the advantages and disadvantages of different dealiasing methods. In this regard, throughout the analysis we

have tried to keep the structure of the LES configuration as simple and canonical as possible, to remove the effect of other add-on complexities. Additional complications might arise when considering additional physics; here we discuss the potential effect that these different dealiasing methods could have on them. One of such elements of added complexity is for example the use of more sophisticated subgrid scale models based on dynamic approaches to determine the values of the Smagorinsky constant (Germano et al., 1991; Bou-Zeid et al., 2005). In most of these advanced subgrid models, information from the small-scale turbulent eddies is used to determine the evolution of the subgrid constant. However, in both the FT and FS method, the small turbulent scales are severely affected and hence use of dynamic subgrid models could be severely hampered unless these are accordingly modified and adjusted, maybe via filtering at larger scales than the usual grid scale. Another element of added complexity consists in using more realistic atmospheric forcing, considering for example the effect of the Coriolis force with flow rotation as a function of height and velocity magnitude. In this case, we hypothesize that the FT method could lead to stronger influences on the resultant flow field as this dealiasing technique not-only affects the distribution of energy in the small turbulent scales, but also in the large scales (as apparent from Fig. 3), being these later ones potentially more affected by the Coriolis force. This represents a strong non-linear effect, that is hard to quantify and hence further testing, including realistic forcing with a geostrophic wind and Coriolis force would be required to better quantify these effects. Also often in LES studies of atmospheric flows one is interested in including an accurate representation of scalar transport (passive/active). In this case the differential equations don't include a pressure term and hence most of the computational cost is linked to the evaluation of the convective term. As a result, the benefit of using alternative, cheaper dealiasing techniques (FT or FS) will be even more advantageous, yet the total gain is not trivial to evaluate *a priori*, and the effect on the scalar fields should also be further evaluated.

In general, we believe that it is not fair to advocate for one or other dealiasing method based on the results of this analysis. Note that the goal of this work is to provide an objective analysis of the advantages and limitations that the different methods provide, letting the readers the ultimate responsibility to choose the option that will adjust better to their application. For example, while having exact dealiasing (3/2-rule) might be better in studies focusing on turbulence and dispersion, one might be well-off using a simpler and faster dealiasing scheme to run the traditionally expensive warm-up runs, or to evaluate surface drag in flow over urban and vegetation canopies, where most of the surface force is due to pressure differences (Patton et al.,2016)."

**Specific responses**

**Major comments:**

1. **Reviewer comment:** *The authors perspective on the FT method is a bit different from mine, and to be honest, I am not sure which one is the most prevalent in the community. I have always considered the FT method (which is also known as the 2/3 rule) as a slightly different implementation of the exact 3/2 rule, but with a reduced number of effective grid points (actually $N_{eff} = 2N/3$ instead of N). In my view, performing a calculation with $3N/2$ points using the FT method should be identical (in simulation results but maybe not in computational cost) to a simulation with N points using the 3/2-rule, no? I also think the that simulations using the FT method should actually report grid resolution based on $N_{eff}$ and not on N (i.e. the grid spacing should be $\Delta x_{eff} = 3\Delta x/2$ instead of $\Delta x$), but that is usually not done. Maybe the authors can comment on this?*

**Authors response:** This is a very important point in which we fully agree with the reviewer. Running a simulation with $3N/2$ points using the FT method will not provide the same fluid-flow results (physics) than a simulation with N points using the 3/2-rule. The FT method is equivalent of using a coarser grid for the convective term. Therefore, some spurious oscillations appear in the flow field as can be seen in figure 6 of the paper, and indicated therein. In addition, as mentioned also by the reviewer, there will be an important difference in computational cost. For the sake of discussion, we have run a comparison test case with a grid of $128^3$ using the 3/2-rule, and another case with a grid of $192, 192, 128$ (so $3N/2$ points in the horizontal direction). Results show that the latter is about 1.65 times more expensive than the former and the physical results are not the same or equivalent either. Specifically, figure 1 shows the instantaneous stream-wise velocity perturbation where some spurious oscillations appear in the flow field. In addition, figure 2 presents the velocity profile, velocity gradient, and variance profiles. The velocity profile obtained with the FT method shows an acceleration of the flow compare to the 3/2-rule, yielding a increase of MKE of 5%. The velocity gradient exhibits a large departure for the FT method at height between $0.2 < z/zi < 0.8$. Similarly, the FT methods yields larger variances throughout the domain.

[Figure]

Figure 1: Instantaneous stream-wise velocity perturbation $u'(x, y, z, t) = u(x, y, z, t) - \bar{u}(x, y, z)$ at $z/z_i = 0.054$ for the $128^3$-simulation with 3/2-rule (a), and the $192, 192, 128$-simulation with FT (b).

2. **Reviewer comment:** *I am surprised by the high share of the cost carried by the convective term compared to a very low share carried by the Poisson solver. Is this something that is specific to the pencil decomposition parallelization? It may be the case that padding with the 3/2 rule is not as simple in a pencil decomposition approach given that one usually needs to pad the entire 2D "horizontal" wave-number space? Maybe some details of the padding in the context of pencil decomposition would be useful. In addition, is it possible that the Poisson solver is faster when pencil decomposition is used?*

[Figure]

Figure 2: Plots of the non-dimensional mean stream-wise velocity profile (a), the mean stream-wise velocity gradient (b), and the non-dimensional variances (c/d/e). The lines represent the $128^3$-simulation with 3/2-rule in blue line and, the $192, 192, 128$-simulation with FT in orange line.

**Authors response:** Indeed. The cost breakdown for the resolution of the convective term and the Poisson solver is also influenced by the pencil decomposition. In treating the convective term with the pencil decomposition the communication cost increases with respect to the traditional slice parallelization. In this case a total of nine transpositions are needed to compute the convective term, significantly increasing the computational cost.

On the other hand, the Poisson solver becomes faster when using the pencil decomposition in comparison to the slice parallelization. Note that in the pseudo-spectral method the horizontal directions ($x$ and $y$) are treated in Fourier space and only the vertical direction ($z$) remains in physical space, therefore each mode in $k_x$ and $k_y$ become independent of each other. In this case the system of equations, originally of size $n_x \times n_y \times n_z$ becomes $n_x \times n_y$ systems of $n_z$ equations, making each vertical line in the domain independent. The pencil decomposition can take full advantage of this fact, making the resolution of the Poisson equation faster. Specifically, once the domain is transposed in the $Z$-pencil (square pipe aligned with the $z$-coordinate), the process of solving each of the $n_x \times n_y$ systems does not require any communication, making it very efficient, and limiting its cost to the transposition between the different pencils.

**Changes in manuscript:** We realize that this is an important detail that should be also mentioned in the manuscript. For this reason we have included a couple of lines in section 3.3.

The text now reads as:"In addition, it is important to note that the low computational cost of the Poisson solver is related to the the use of the pencil decomposition, which takes full advantage the pseudo-spectral approach. Specifically, the $Z$-pencil combines with the horizontal treatment of the derivatives to make the implementation of the solver very efficient."

3. **Reviewer comment:** *The quantification of computation cost in section 4.1 is fine. Regarding the results in section 4.2, I wonder if the question could be posed the other way around. In my view, maybe the most relevant question would be: "given a set computational cost, do I get better results (a) using the 3/2 rule or (b) running a finer grid using a less expensive dealiasing?". I know the experiment is not easy to design (because one needs to estimate the computational cost ahead of time), but that seems to be the more important question to be answered. If I save 30% on dealiasing and spend it on more resolution, do I get better results? From my comment above, I would expect that the FT method is almost equivalent to the 3/2 rule (if smartly coded). How about the FS?*

**Authors response:** This is a very interesting point that is being raised. It is true that *a-priory*, as the reviewer mentions, one could compute the associated computational gain linked to using the FT method and decide using a finer grid that would 'use' the time resources saved in the benefit of resolution. However this is a challenging endeavor for many different reasons. For example, if one was to consider an a-priory 20% gain when using the FT method on a given numerical grid of $128^3$ points, and decided to use a more resolved grid to 're-invest' the saved computational resources, then one would have to use a grid of $\sim 150\,(128 + 0.2 \times 128)$. Unfortunately, this number of grid points will not work well with the FFT given that it's not a power of 2 and hence will induce a slow down of the computations. Accounting for this factor is quite challenging, if not unfeasible, turning the reviewer's suggestion in a very hard challenge. In this regard, also note the response to the first point raised where it is clarified the fact that the 3/2 rule does not provide the exact same results to the case using the 2/3 rule with an equivalently larger grid.

Alternatively, the gain in computational time could be for example invested in running longer simulations, or reduce the time in 'warm-up' configurations.

**Changes in manuscript:** We have included an additional comment related to the possible use of the FT or FS method as precursor simulations in the new discussion section (reported above).

4. **Reviewer comment:** *Regarding the interpretation of Figure 7. I am not convinced the log-law prediction is as good as described in the text (but perhaps I am missing something here?). First, it seems that there is one log-law on top of the solid line that may extend only for 2 or 3 vertical levels (ending around $z/z_i \leq 0.02$). Then there is a second log-law (only in 3/2 and FT cases) that starts*

*around $z/z_i = 0.04$ and goes beyond $z/z_i = 0.1$, This second log-law has the incorrect roughness (if one were to extrapolate it to $u = 0$, it would yield a lower value of $z_0$ than the one imposed on the simulation, I think). There is no clear second log-law in the FS method. I would see this as a concern for the FS method (which is being advocated here), except that not even the exact 3/2 rule has a good log-law (as seen in Bou-Zeid et al, 2005). I am pretty sure this is due to the SGS model adopted here. In any case, if I had to choose between the FS and FT methods based on Figure 7, I would probably go with FT, since it does a reasonable job in the lower 20% of the domain (which is the region of interest in a simulation like this, I guess). Also, the FS method seems more over-dissipative near the wall (on panel b), which is opposite to what is described in the text?*

**Authors response:** We agree with the reviewer that the log-law predictions are far from perfect, yet they match well with those presented in Bou-Zeid et al, (2005) when using the constant Smagorinsky coefficient (contrasted on a side work). This means that the deviations from the theoretical log-law are mostly due to the SGS model, as the reviewer suggests. These deviations are more prominent in the FS method whereas the FT shows excellent agreement with predictions from the 3/2-rule in the surface layer.

In regards to the second comment, we believe that there was a misunderstanding, given that the authors comments in the manuscript referred to the upper region of the BL, while the reviewer is referring to the surface layer region.

**Changes in manuscript:** We have now clarified both issues in the revised version of the manuscript. The interpretation of the figure 7 now reads:
"The horizontally- and temporally-averaged velocity profiles are characterized by an approximately logarithmic behavior within the surface-layer ($z \approx 0.15 z_i$, as apparent from Fig. 7, where results are illustrated for the three resolutions: $128^3$, $192^3$, and $256^3$). For the $128^3$ case, the observed departure from the logarithmic profile for the 3/2-rule case is in excellent agreement with results from previous literature for this particular SGS model (Port-Agel et al., 2000; Bou-Zeid et al., 2005). When using the FT method the agreement of the averaged velocity profile with the corresponding 3/2-rule profiles improves with increasing resolution. While in the $128^3$ case a good estimation of the logarithmic flow is obtained at the surface layer, there is a large acceleration of the flow further above. This overshoot does not occur for the higher resolution runs. When using the FS method, the mean velocity magnitude is consistently over-predicted throughout the domain, and the situation does not improve with increasing resolution (the overshoot is up to 7.5% for the $128^3$, 8.5% for the $192^3$ and 7% for the $256^3$ run). Further comparing the results obtained by the FS and FT method with those obtained with the 3/2-rule, it is clear that while the FS method presents a generalized overestimation of the velocity with a an overall good logarithmic trend, the FT method presents a better adjustment in the surface layer with larger departures from the logarithmic regime on the upper domain region that get reduced with increasing numerical resolution. The mean kinetic energy of the system is overestimated by $\approx +2\%$ and $\approx +12\%$ by the FT and FS methods, when compared to that of runs using the 3/2-rule in the $256^3$ case. Overall, the mean kinetic energy is larger for the FT and FS cases, when compared to the 3/2-rule case, even at the highest of the considered resolutions ($\approx +2\%$ and $\approx +12\%$ by the FT and FS methods for the $256^3$ case). Such behavior can be related to the low-pass filtering operation that is performed in the near-wall regions, which tends to reduce resolved turbulent stresses in the near-wall region, resulting in a higher mass flux for the considered flow system. This is more apparent for the low resolution cases.

Mean velocity gradient profiles ($\Phi_m = \kappa \frac{z}{u_*} \partial_z \langle U \rangle_{xy}(z)$) are also featured in Figure 7 (d, e,

f). Profiles at each of the considered resolutions present a large overshoot near the surface, which is a well known problem in LES of wall-bounded flows and has been extensively discussed in the literature (Bou-Zeid et al., 2005; Brasseur and Wei, 2010; Lu and Port-Agel, 2013). In comparing the results between the FS and FT method with the 3/2-rule, it can be observed that there are stronger gradients in the mean velocity profile within the surface layer when using the FS method. This leads to the observed shift in the mean velocity profile. Conversely, when using the FT method, departures are of oscillatory nature, leading to less pronounced variations in the mean velocity profile when compared to the reference ones (the 3/2-rule cases). This behavior is consistently found across the considered resolutions, but the situation ameliorates as resolution is increased (*i.e.* weaker departures)."

5. **Reviewer comment:** *Regarding Figure 9. I do not agree that the filtering only affects the small-scale end of the spectra. For the FT case, it seems clear to me that there is significant damping of the large scales as well. This is related to the underestimation in the variance of the streamwise velocity seen in Figure 8. This is worrisome, and probably related to the fact that the true resolution here ($N_{eff} = 85$ points) is too coarse to model the ABL. I wonder if this situation would persist in the $256^3$ simulations?*

**Authors response:** Thank you very much, this is a very interesting and important point. The underestimation of the variance of the streamwise velocity did not persist in the $256^3$ simulations. In addition, to clarify the effect of the FT and FS methods on the spectra, we have developed an additional analysis using the spectra presented in the paper. Although the effect of the FT and FS methods on the small scales can be clearly observed on the spectra, their effect on the large scales cannot be directly assessed from the figure, as pointed out by the reviewer. To compute a direct comparison scale by scale, the following ratio was used for the $128^3$, $192^3$, and $256^3$-simulations,

$$\rho^{XX}(k) = \frac{E_{u,k}^{XX} - E_{u,k}^{3/2}}{E_{u,k}^{3/2}} \tag{1}$$

where $E_{u,k}$ denotes the power spectral density of the $u$ velocity component at wavenumber $k$, $XX$ stands for the dealiasing method FT or FS. Hence, if $\rho(k) < 0$ energy is removed at that scale, and if $\rho(k) > 0$ energy is added at that scale. Figure 3 presents the ratio $\rho(k)$ for both methods where it can be observed that the effect of FT methods is very large at all scales. The large scales ($0 \leq k/k_{max} \leq 0.2$) are affected with a reduction of energy of $\sim$25%. The mid-range scales ($0.2 \leq k/k_{max} \leq 0.6$), corresponding to the inertial sub-range, exhibit an overestimation of their energy of about $\sim$50% on average. Therefore, this method redistributes the energy of the small scales into the inertial sub-range scales. On the contrary, in the FS method, the energy from the filtered small-scales is redistributed more or less uniformly throughout with an averaged overall variation of less the 13%.

**Changes in manuscript:** We have added figure 3 and its interpretation in the manuscript. This now reads as: "Although the effect of the FT and FS methods on the small scale can be clearly observed in figure 9, their effect on the large scales also needs to be quantified. To compute a direct comparison scale by scale, the following ratio was used (equation 1) for the $128^3$, $192^3$, and $256^3$-simulations,

$$\rho^{XX}(k) = \frac{E_{u,k}^{XX} - E_{u,k}^{3/2}}{E_{u,k}^{3/2}} \tag{2}$$

[Figure]

Figure 3: Effect of the FT (a), and the FS (b) methods of the stream-wise spectra of the stream-wise velocity compare to the 3/2-rule. The solid line represent the average value and the shaded area represent the extreme values. The resolutions are: $128^3$ in blue dot-dashed line, $192^3$ in red dotted line, and $256^3$ in purple dashed line.

where $E_{u,k}$ denotes the power spectral density of the $u$ velocity component at wavenumber $k$, $XX$ stands for the dealiasing method FT or FS. If $\rho(k) < 0$ the energy density at that given wavenumber ($k$) is less than the corresponding one for the run using the 3/2 rule, viceversa if $\rho(k) > 0$. Figure 3 presents the ratio $\rho(k)$ for both methods.

When using the FT method, energy at the low wavenumbers is underpredicted, whereas energy at the large wavenumbers is overpredicted. Departures are in general larger with decreasing resolution, with an excess of up to $100\%$ for the $128^3$-simulations in the wavenumber range close to the cutoff wavenumber. On the contrary, when using the FS method, the energy from the filtered (dealiased) small-scales is redistributed quasi-uniformly throughout the spectra with an averaged overall variation of less than $13\%$."

6. **Reviewer comment:** *I see two facts that could benefit from more discussion in the manuscript (maybe in the conclusions):*
*(a) The importance of reducing computational cost in the advection term for simulations that use more sophisticated SGS models (maybe this can be brought up again in the conclusions?)*
*(b) The consequence for including one or many additional scalar fields (temperature, water vapor, etc.). These also require dealising and do not require additional pressure solvers and/or very expensive SGS models. I would probably anticipate that in simulations with several scalars, the savings would be significantly larger. Is that correct?*

**Authors response:** Thank you, (a) is a great point. In this regard, we have now run some additional simulations using the dynamic Smagorinsky model. Results are illustrated in Figure 4. In this figure, it can be observed that the dynamic model fails to compute the Smagorinsky constant when using either the FS or the FT methods. In addition, the FT methods strongly suppresses the turbulence, resulting in the lamieraizition of the flow. Alternatively, the consequences of using the FS method are less dramatic, although the flow also exhibits a large acceleration at the top of the domain. As mentioned by the reviewer, these results are not surprising given that the dynamic models are using a relation between the small scales to compute the Smagorinsky constant. Therefore, the FT and FS methods cannot be used

[Figure]

Figure 4: Profiles of the horizontal velocity and the Smagorinsky constant for the three deasliasing methods at a resolution of $128^3$.

with dynamic SGS models unless the dynamic models are properly adjusted with filtering at scales larger than those affected by the truncation or smoothing. Therefore, it is impossible to *re-invest* the computation gain due to the FT or FS methods into a more sophisticate SGS model (at lease not a dynamic Smagorinsky-type model from Germano (1991)).

In relation to the second comment, adding extra scalars will decrease the cost of the momentum solver with respect to the total cost of the simulation. Each scalar field is advanced in time using an advection-diffusion equation that also requires dealiasing. However, the cost of the dealiasing of the latter equation is less expensive than the NS equations (especially in rotational form). To summarize, each dealiasing operation becomes less expensive with the FT or FS method, and the total gain will be more important as more operations are required. However, the total gain is not trivial to evaluate *a priori*.

**Changes in manuscript:** We agree that both issues are relevant, and hence we have included some additional comment in the new discussion section (reported above).

**Minor comments:**

1. **Reviewer comment:** *Page 4, Line 33 - please check that the $N^3$ term in the cost is correct here.*

   **Authors response:** Thank you for pointing out this mistake. This cost should not have a $N^3$ but only $N$.

2. **Reviewer comment:** *Page 5, Line 10 - delta implicit does not really correspond to a top-hat filter. The properties of an implicit filter are tied to the discretization scheme. As an example, for a true 3D spectral code the implicit filter is a spectral cutoff filter.*

   **Authors response:** Thank you this is a good point. This has been corrected in the text.

---

## Author Comment (AC2) · 2 May 2018

**Title:** *Comparison of dealiasing schemes in large-eddy simulation of neutrally-stratified atmospheric boundary-layer type flows*

**Authors:** *Fabien Margairaz, Marco G. Giometto, Marc B. Parlange, and Marc Calaf*

**Submitted to:** Geosci. Model Dev. (gmd-2017-272)

**Reviewer:** Referee #2

**Date:** May 1, 2018

**General response**

**Reviewer general comment:** *The paper compares various approaches for the dealiasing of the non-linear terms in large eddy simulation of atmospheric flows. Given that spectral methods are widely used in studying such flows under idealized conditions since they offer higher speed and better accuracy, the general theme is of interest to GMD readers. The paper does a good job in presenting the fundamentals of the problem, the proposed solutions, and how they compare when implemented in an actual code. But major revisions are needed. In particular, the study is a valuable comparison of the methods that is not available (to the best of my knowledge) in the literature and the authors should not try to conclude that one is more optimal than the others. They can simply present their findings and let the users determine which method is suitable for their needs.*

**Authors response:** We thank the reviewer for his/her valuable comments and for recommending publication of the manuscript. In this regard we have taken good note of the reviewer comments, which have been addressed in detail below.

Based on the comments from both reviewers, we have added a new discussion section to improve the readability of the manuscript. We have also improved the results section by running additional simulations at the resolutions of $192, 192, 128$ and $192^3$, as well as running the $256^3$-case long enough to be able to report converged statistics. As a result Figures 7, 8 and 9 have been updated, and a new figure (Figure 10) has been included, which is discussed at the end of the results section.

**Changes in manuscript:** The new discussion section is reported here:

"In the development of this manuscript, focus has been directed to the study of the advantages and disadvantages of different dealiasing methods. In this regard, throughout the analysis we have tried to keep the structure of the LES configuration as simple and canonical as possible, to remove the effect of other add-on complexities. Additional complications might arise when considering additional physics; here we discuss the potential effect that these different dealiasing methods could have on them. One of such elements of added complexity is for example the use of more sophisticated subgrid scale models based on dynamic approaches to determine the values of the Smagorinsky constant (Germano et al., 1991; Bou-Zeid et al., 2005). In most of these

advanced subgrid models, information from the small-scale turbulent eddies is used to determine the evolution of the subgrid constant. However, in both the FT and FS method, the small turbulent scales are severely affected and hence use of dynamic subgrid models could be severely hampered unless these are accordingly modified and adjusted, maybe via filtering at larger scales than the usual grid scale. Another element of added complexity consists in using more realistic atmospheric forcing, considering for example the effect of the Coriolis force with flow rotation as a function of height and velocity magnitude. In this case, we hypothesize that the FT method could lead to stronger influences on the resultant flow field as this dealiasing technique not-only affects the distribution of energy in the small turbulent scales, but also in the large scales (as apparent from Fig. 2), being these later ones potentially more affected by the Coriolis force. This represents a strong non-linear effect, that is hard to quantify and hence further testing, including realistic forcing with a geostrophic wind and Coriolis force would be required to better quantify these effects. Also often in LES studies of atmospheric flows one is interested in including an accurate representation of scalar transport (passive/active). In this case the differential equations don't include a pressure term and hence most of the computational cost is linked to the evaluation of the convective term. As a result, the benefit of using alternative, cheaper dealiasing techniques (FT or FS) will be even more advantageous, yet the total gain is not trivial to evaluate *a priori*, and the effect on the scalar fields should also be further evaluated.

In general, we believe that it is not fair to advocate for one or other dealiasing method based on the results of this analysis. Note that the goal of this work is to provide an objective analysis of the advantages and limitations that the different methods provide, letting the readers the ultimate responsibility to choose the option that will adjust better to their application. For example, while having exact dealiasing (3/2-rule) might be better in studies focusing on turbulence and dispersion, one might be well-off using a simpler and faster dealiasing scheme to run the traditionally expensive warm-up runs, or to evaluate surface drag in flow over urban and vegetation canopies, where most of the surface force is due to pressure differences (Patton et al.,2016)."

**Specific responses**

**Major comments:**

1. **Reviewer comment:** *While the FS method seem to be giving an acceptable performance as the authors argue, I wonder whether the ABL LES community should be going in a direction of saving computing time rather than maximizing the accuracy of the computation. We push for higher resolution to gain better accuracy and, with increasing computing power, I wonder whether a 20% drop in simulation time is worth it. We use dynamic SGS models that increase the computing time by 20% all the time. The plots in Fig 7 do not indicate that the FS method is as good as the 3/2 method. So in general I think the authors should not focus on the conclusion that the FS method is a good surrogate. They should present the information and findings, which will help modelers decide on the trade offs they want (on my end this convinces me that using a 3/2 method is indeed worth it.).*

   **Authors response:** Thanks, we indeed agree with the reviewer's point. We believe that gaining a 20% in computational time can be of interest in certain occasions, for example during warm up periods. However, as the reviewer mentions the strength of this manuscript should reside on conveying the facts of using different dealiasing methods, and allowing the corresponding end users to decide what's best for them according to their application.

**Changes in manuscript:** To clarify this point we have added a discussion section (reported above) and rewritten the conclusion section. The new conclusion section emphasize the trade offs of each method. The modifications to conclusion reported here:

"The Fourier-based pseudo-spectral collocation method (Orszag, 1970; Orszag and Pao, 1975; Canuto et al., 2006) remains the preferred "work-horse" in simulations of wall-bounded flows over horizontally periodic regular domains, which is often used in conjunction with a centered finite-difference or Chebychev polynomial expansions in the vertical direction (Shah and Bou-Zeid,2014; Moeng and Sullivan, 2015). This approach is often used because of the high-order accuracy and the intrinsic efficiency of the fast-Fourier-transform algorithm (Cooley and Tukey, 1965; Frigo and Johnson, 2005). In this technique, aliasing that arises when evaluating the quadratic non-linear term in the NS equations can severely deteriorate the quality of the solution and hence need to be treated adequately. In this work a performance/cost analysis has been developed for three well-accepted dealiasing techniques (3/2-rule, FT and FS) to evaluate the corresponding advantages and limitations. The 3/2-rule requires a computationally expensive padding and truncation operation, while the FT and FS methods provide an approximate dealiasing by low-pass filtering the signal over the available wavenumbers, which comes at a reduced cost.

The presented results show compelling evidence of the benefits of these methods as well as some of their drawbacks. The advantage of using the FT or the FS approximate dealiasing methods is their reduced computational cost ($\sim$15% for the $128^3$ case, $\sim$25% for the $256^3$ case), with an increased gain as the numerical resolution is increased. Regarding the flow statistics, results illustrate that both, the FT and the FS methods, yield less accurate results when compared to those obtained with the traditional 3/2-rule, as one could expect.

Specifically, results illustrate that both the FT and FS methods over-dissipate the turbulent motions in the near wall region, yielding an overall higher mass flux when compared to the reference one (3/2-rule). Regarding the variances, results illustrate modest errors in the surface-layer, with local departures in general below 10% of the reference value across the considered resolutions. The observed departures in terms of mass flux and velocity variances tend to reduce with increasing resolution. Analysis of the streamwise velocity spectra has also shown that the FT method redistributes unevenly the energy across the available wavenumbers, leading to an over-estimation of the energy of some scales by up to 100%. Contrary, the FS methods redistributes the energy evenly, yielding a modest +13% energy magnitude throughout the available wavenumbers. Compared to the 3/2-rule, these differences in flow statistics are the result of the sharp low-pass filter applied in the FT method and the smooth filter that characterizes the FS method."

2. **Reviewer comment:** *How do the FT and FS method influence the potential use of dynamic models that require good accuracy on the smallest resolved scales? If as the spectra show they damp these scales, than that would preclude using dynamic models and would be a significant disadvantage of FT and FS. The authors have in their code some dynamic models, they could perform the dynamic computations while still using the Static Smagorinsky (compute a dynamic Cs but dont use it).*

**Authors response:** Thank you, this is a great point. In this regard, we have now run some additional simulations using the dynamic Smagorinsky model. Results are illustrated in Figure 1. In this figure, it can be observed that the dynamic model fails to compute the Smagorinsky constant when using either the FS or the FT methods as traditionally implemented. In this case the FT method strongly suppresses turbulence, resulting in the laminarization of the flow. Alternatively, while the consequences of using the FS method are less dramatic,

the flow also exhibits a large acceleration at the top of the domain. As mentioned by the reviewer, these results are not surprising given that the dynamic models are using a relation between the small scales to compute the Smagorinsky constant. We believe that these could though be slightly improved by using information from scales larger than the traditional filtering scale, if the reader was really interested. Yet this reamins outside the scope of this manuscript. Therefore, it is advisable not to use the FT and FS methods with dynamic SGS

[Figure]

Figure 1: Profiles of the horizontal velocity and the Smagorinsky constant for the three deasliasing methods at a resolution of $128^3$.

models. In this regard, we have added additional text in the new discussion section that relates to the use of dynamic models and the fact that they probably require some modification to run with the FT and FS methods.

**Changes in manuscript:** A discussion related to this comment has been added to the discussion section (see new discussion and conclusion section in comment #1).

3. **Reviewer comment:** *Fig 8d and the associate sentence "Interestingly, results of the vertical flux (or stress, resolved and SGS) of stream-wise momentum (figure 8(d)) illustrate a good agreement between the different scenarios." The authors should be careful in this interpretation. The constant pressure gradient forcing requires and forces the stress profile to be linear. Regardless of how turbulence ends up looking like the turbulent fluxes have to adjust to balance the mean $\partial P/\partial x$. What this figure indicates is that the SGS fraction is not strongly affected by the choice of dealiasing method, which is a good thing.*

**Authors response:** Thank you for bringing this to our attention. As mentioned above the results section has been adjusted where this is taken care of.

4. **Reviewer comment:** *Figure 9, and more generally: I would have liked to see a direct comparison of*

*the largest scales (by filtering all simulations at $n\Delta$ , where n correspond to start of the damping or cutoff in figure 1) to see if the differences are only on the smallest scales or not (although given the mean velocity profiles, I suspect they are not).*

**Authors response:** Thank you very much, this is a very interesting and important point. In order to answer this comment and clarify the effect of the FT and FS methods on the spectra, we have developed an additional analysis using the spectra presented in the paper. Although the effect of the FT and FS methods on the small scales can be clearly observed on the spectra, their effect on the large scales cannot be directly assessed from the figure, as pointed out by the reviewer. To compute a direct comparison scale by scale, the following ratio was used for the $128^3$, $192^3$, and $256^3$-simulations,

$$\rho^{XX}(k) = \frac{E_{u,k}^{XX} - E_{u,k}^{3/2}}{E_{u,k}^{3/2}} \tag{1}$$

where $E_{u,k}$ denotes the power spectral density of the $u$ velocity component at wavenumber $k$, $XX$ stands for the dealiasing method FT or FS. Hence, if $\rho(k) < 0$ energy is removed at that scale, and if $\rho(k) > 0$ energy is added at that scale. Figure 2 presents the ratio $\rho(k)$ for both methods where it can be observed that the effect of FT methods is very large at all scales. The large scales ($0 \leq k/k_{max} \leq 0.2$) are affected with a reduction of energy of $\sim$25%. The mid-range scales ($0.2 \leq k/k_{max} \leq 0.6$), corresponding to the inertial sub-range, exhibit an overestimation of their energy of about $\sim$50% on average. Therefore, this method redistributes the energy of the small scales into the inertial sub-range scales. On the contrary, in the FS method, the energy from the filtered small-scales is redistributed more or less uniformly throughout with an averaged overall variation of less the 13%.

**Changes in manuscript:** We have added figure 2 and its interpretation in the manuscript. This now reads as: "Although the effect of the FT and FS methods on the small scale can be clearly observed in figure 9, their effect on the large scales also needs to be quantified. To compute a direct comparison scale by scale, the following ratio was used (equation 2) for the $128^3$, $192^3$, and $256^3$-simulations,

$$\rho^{XX}(k) = \frac{E_{u,k}^{XX} - E_{u,k}^{3/2}}{E_{u,k}^{3/2}} \tag{2}$$

where $E_{u,k}$ denotes the power spectral density of the $u$ velocity component at wavenumber $k$, $XX$ stands for the dealiasing method FT or FS. If $\rho(k) < 0$ the energy density at that given wavenumber ($k$) is less than the corresponding one for the run using the $3/2$ rule, viceversa if $\rho(k) > 0$. Figure 2 presents the ratio $\rho(k)$ for both methods.

When using the FT method, energy at the low wavenumbers is underpredicted, whereas energy at the large wavenumbers is overpredicted. Departures are in general larger with decreasing resolution, with an excess of up to 100% for the $128^3$-simulations in the wavenumber range close to the cutoff wavenumber. On the contrary, when using the FS method, the energy from the filtered (dealiased) small-scales is redistributed quasi-uniformly throughout the spectra with an averaged overall variation of less than 13%."

[Figure]

Figure 2: Effect of the FT (a), and the FS (b) methods of the stream-wise spectra of the stream-wise velocity compare to the 3/2-rule. The solid line represent the average value and the shaded area represent the extreme values. The resolutions are: $128^3$ in blue dot-dashed line, $192^3$ in red dotted line, and $256^3$ in purple dashed line.

**Minor comments:**

1. **Reviewer comment:** *Title is long and too descriptive: how about replacing the wordy "atmospheric boundary- layer type flows" with "atmospheric flows". One in fact could foresee using such methods for cloud resolving LES outside the ABL. Same on last line of abstract: why restrict the applications only to ABL flows?*

    **Authors response:** Thank you for this comment. We have adapted these very interesting suggestions. The new title is included below, as well as the new abstract.

    **Changes in manuscript:** Title: "Comparison of dealiasing schemes in large-eddy simulation of neutrally-stratified atmospheric flows"

    Abstract: "Aliasing errors arise in the multiplication of partial sums, such as those encountered when numerically solving the Navier-Stokes equations, and can be detrimental to the accuracy of a numerical solution. In this work, a performance/cost analysis is proposed for widely-used dealiasing schemes in large-eddy simulation, focusing on a neutrally-stratified, pressure-driven atmospheric boundary-layer flow. Specifically, the exact 3/2 rule, the Fourier truncation method, and a high order Fourier smoothing method are inter-compared. Tests are performed within a newly developed mixed pseudo-spectral collocation - finite differences large-eddy simulation code, parallelized using a two-dimensional pencil decomposition. A series of simulations are performed at varying resolution and key flow statistics are

inter-compared among the considered runs and dealiasing schemes. Both the Fourier Truncation and the Fourier Smoothing method correctly predict basic statistics. However, they both prove to provide less accurate flow statistics when compared to the traditional 3/2-rule. The accuracy of the methods is dependent of the resolution. The biggest advantage of both of these methods against the exact 3/2-rule is a notable reduction in computational cost with an overall reduction of $15\%$ for a resolution of $128^3$, $17\%$ for $192^3$ and $21\%$ for $256^3$."

2. **Reviewer comment:** *Abstract line 3: better to replace "integrating" by "time advancing"*

   **Authors response:** Thank you for pointing this out. The abstract has been significantly changed (see comment 1).

3. **Reviewer comment:** *Abstract lines 4-5: not sure what is meant by "This is of special relevance when using high order schemes." Spectral schemes are always "high order"*

   **Authors response:** Thank you. We realize that this sentence was miss-leading. In the manuscript, we were referring to the schemes used for the 3rd dimension and for the time advancement. The abstract has been significantly changed (see comment 1).

4. **Reviewer comment:** *First 3 lines (19-21) of introduction. In fact the # of grid points in LES has not been following Moores law. See https://doi.org/10.1017/jfm.2014.616 . This shows that the LES community has not been taking full advantage of increasing computing power to improve model accuracy, which I think is a remiss.*

   **Authors response:** Thank you for bringing this point to our attention.

   **Changes in manuscript:** We have added this comment in the introduction: "Despite this progress, high resolution simulations effectively exploiting current hardware and software capabilities (i.e., following Moore's law) are challenging as they require significant computational resources, which most research groups do not dispose of (Bou-Zeid, 2014). As a result, methods that aim at reducing computational requirements while preserving numerical accuracy are still of great interest."

5. **Reviewer comment:** *Page 2, line 2. Add comma after "With increasing computer power". I think there are a few other missing commas after introductory phrases.*

   **Authors response:** Thank you for pointing out these omissions. These have been amended.

6. **Reviewer comment:** *Correct wall bounded flow to wall-bounded flow on page 2 line 7.*

   **Authors response:** Thank you for pointing this out. This has been corrected.

7. **Reviewer comment:** *Page 2, Lines 30-32: authors talk about the need to expand the grid onto 3/2N and then say "As a result, due to the non-linear dependence on N" This example make it sound linear. I think they are referring to the non-linearity of the FFT cost with N, which they explain later. Clarify?*

   **Authors response:** Thank you for bringing this up. We realize that the sentence was confusing, and hence we have changed it in the text. In the manuscript, we are indeed referring to the cost of the FFT, given that the size of the expended grid is linear with $N$.

   **Changes in manuscript:** We have made the following modifications: "As a result, the computational burden introduced by these methods is high, mainly due to the non-linear increase of the cost of the fast Fourier transform algorithm (such as the one implemented in

the FFTW library). Additionally, this cost rises more rapidly when the Fourier transform is performed in higher dimensions. Therefore, the treatment of aliasing errors severely limits the computational performances of large scale models based on high-order schemes."

8. **Reviewer comment:** *Also maybe they should clarify that if FFTs are used in 2D, the cost rises even more quickly as N rises.*

   **Authors response:** Thank you. We have clarified this point in the text (see comment 7)

9. **Reviewer comment:** *Page 3 line 5 "via a set of LES of fully developed ABL type flows and with a corresponding comparison on the effect in turbulent flow statistics and topology". Convoluted phrase. Simplify.*

   **Authors response:** Good point. We have simplified this phrase.

   **Changes in manuscript:** It now reads as: "In this work, we provide a cost-benefit analysis and a comparison of turbulent flow statistics for the FT and FS dealiasing schemes in comparison to the exact 3/2-rule using a set of LES of fully developed ABL type flows."

10. **Reviewer comment:** *Page 3, line 8 "environmental fluids"*

    **Authors response:** Thank you for pointing out the misspelling.

11. **Reviewer comment:** *Page 4, line 19 "thus discouraging its use in most practical situations" are the authors certain of this statement? The 3/2 rule is use very very widely. If not maybe their paper would be a good reference for modelers to see the advantages of using it.*

    **Authors response:** In this sentence we were originally referring to the phase shift method not to the 3/2 rule. We are well aware that the 3/2-rule is widely used. In this regard we realize that the text was not very clear, and hence we have rewritten the sentence.

    **Changes in manuscript:** It now reads as: "This method has a cost equal to $15N \log_2(N)$ (Canuto et al., 2006), which is even greater than the 3/2-rule (Patterson, 1971; Orszag, 1972), discouraging its use in most practical situations."

12. **Reviewer comment:** *Page 5 line 13, and maybe other places. Referring to the production range as the energy containing range is inaccurate and misleading. The statement "For this technique to be successful, the low-pass filter operation must be performed at a scale smaller than the smallest energy containing scale, deep in the inertial sub-range according to Kolmogorovs hypothesis (Kolmogorov, 1968; Piomelli, 1999)." For example makes no sense if that jargon is used. Please fix and use energy production range instead.*

    **Authors response:** Thanks, this is a very good point. In this regard we have revised the manuscript and changed this reference for 'energy production range' as indicated by the reviewer.

13. **Reviewer comment:** *Eq 8, the $\tau$ should have a d superscript if the trace is already in $p*$ as the authors write.*

    **Authors response:** Thank you for pointing this out. This has been corrected.

14. **Reviewer comment:** *Page 6, line 18: the log law is not inviscid since it is derived from matching the viscous sublayer and the outer layer. If the wall is smooth for example $z_0$ depends on viscosity.*

    **Authors response:** Thank you for pointing this out. We have modified the text to avoid the

miss-understanding.

**Changes in manuscript:** The text now reads as: "Note that the molecular viscous term has been neglected within the flow. However, the effect of the molecular viscosity at the surface is modeled using the logarithmic law, where the surface drag is parameterized through the surface roughness."

15. **Reviewer comment:** *Page 6 line 25: what does "module" mean? Do they mean modulus?*

    **Authors response:** Indeed. We have clarified this in the manuscript.

16. **Reviewer comment:** *Should equation 12 include $f_i$ to be consistent ?*

    **Authors response:** Thanks, this has been corrected.

17. **Reviewer comment:** *Page 8, lines 10-15: Authors should clarify this is with the baseline 3/2 dealiasing I presume. Also how does the parallelization method impact these numbers?*

    **Authors response:** Thank you for pointing this out, we used the 3/2 as a baseline. In addition, an other comment on this regard was also made by an other reviewer, and hence we have included some clarification in the manuscript.

    Here is our response to the question regarding the parallelization and the pressure solver. We have noticed that the cost breakdown for the resolution of the convective term and the Poisson solver is also influenced by the pencil decomposition.

    When treating the convective term with the pencil decomposition, the communication cost increases with respect to the traditional slice parallelization. In this case, a total of nine transpositions are needed to compute the convective term, significantly increasing the computational cost.

    Opposite, the Poisson solver becomes faster when using the pencil decomposition in comparison to the slice parallelization. Note that in the pseudo-spectral method the horizontal directions ($x$ and $y$) are treated in Fourier space and only the vertical direction ($z$) remains in physical space, therefore each mode in $k_x$ and $k_y$ become independent of each other. In this case the system of equations, originally of size $n_x \times n_y \times n_z$ becomes $n_x \times n_y$ systems of $n_z$ equations, making each vertical line in the domain independent. The pencil decomposition can take full advantage of this fact, making the resolution of the Poisson equation faster. Specifically, once the domain is transposed in the $Z$-pencil (square pipe aligned with the $z$-coordinate), the process of solving each of the $n_x \times n_y$ systems does not require any communication, making it very efficient, and limiting its cost to the transposition between the different pencils.

    **Changes in manuscript:** We realize that this is an important detail that should be also mentioned in the manuscript. For this reason we have included a couple of lines in section 3.3.

    The text now reads as: "In addition, it is important to note that the low computational cost of the Poisson solver is related to the the use of the pencil decomposition, which takes full advantage the pseudo-spectral approach. Specifically, the $Z$-pencil combines with the horizontal treatment of the derivatives to make the implementation of the solver very efficient."

18. **Reviewer comment:** *Page 10, lines 1-2: The $z_0$ they impose is 1cm, which corresponds more to a grass field than to a sparse forest of to a farmland. I suggest they check Brutsaerts books rather than to Stull for $z_0$.*

**Authors response:** We realized that there was a mistake in the actual value of $z_0$. This one is actually of 0.1m, which corresponds to a sparse forest according to Stull and Brutsaert. We added the reference to Brutsaert's book in the manuscript.

19. **Reviewer comment:** *Figure 5 and other are difficult to read. Why not use colors for the online version (Color is free with EGU, no?)*

    **Authors response:** Because we couldn't find any information in this regard on the publisher web page we decided to go on black & white. If the editor confirms that color figures are free of charge, then we would be happy to change them

20. **Reviewer comment:** *Page 11, lines 9-11: 30% drop in the convective term cost is good but I would not say it is significant. It would only be equivalent to about 20% drop in total computing time (given Fig 2), which would only be equivalent to a 5% reduction in the resolution. So I would remove significantly on line 9.*

    **Authors response:** We have removed the "significantly" in the text. Additional detail in this regard has been provided earlier in comment 17.

21. **Reviewer comment:** *Page 11, lines 10-11: "the predicted computational cost predicted by" remove one of the "predicted".*

    **Authors response:** Thank you for pointing out the repetition.

22. **Reviewer comment:** *Page12, line 2: replace "extend" with "extent"*

    **Authors response:** Thank you for pointing out the misspelling.

23. **Reviewer comment:** *Page12, line 9: correct the misspelling of "stream-wise"*

    **Authors response:** Thank you for pointing out the misspelling.

24. **Reviewer comment:** *Page 12 line 25, and page 14 line 10: "differentiated" is an unclear word. Please remove and clarify the two sentences.*

    **Authors response:** Thank you for pointing out these two sentences. The discussion of the results have been changed and these sentences have been removed.

25. **Reviewer comment:** *Page 14 lines 15-16. There wont be any dispersive stresses in their simulations over homogeneous terrain so why mention them?*

    **Authors response:** In order to avoid any confusion in the discussion, we remove the mention to dispersive stresses in the text